# β-Carotene alleviates substrate inhibition caused by asymmetric cooperativity

Jieren Liao[1,11], Umar F. Shahul Hameed[2,11], Timothy D. Hoffmann [1], Elisabeth Kurze[1], Guangxin Sun[1,3], Wieland Steinchen[4,5], Alessandro Nicoli [6,7], Antonella Di Pizio [6,7], Christina Kuttler [8], Chuankui Song[9], Dragana A. M. Catici[10], Farhah Assaad-Gerbert [1], Thomas Hoffmann [1], Stefan T. Arold [2] & Wilfried G. Schwab [1] ✉

Enzymes are essential catalysts in biological systems. Substrate inhibition, once dismissed, is now observed in 20% of enzymes[1] and is attributed to the formation of an unproductive enzyme-substrate complex, with no structural evidence of unproductivity provided to date[1–6]. This study uncovers the molecular mechanism of substrate inhibition in tobacco glucosyltransferase *Nb*UGT72AY1, which transfers glucose to phenols for plant protection. The peculiarity that β-carotene strongly attenuates the substrate inhibition of *Nb*UGT72AY1, despite being a competitive inhibitor, allows to determine the conformational changes that occur during substrate binding in both active and substrate-inhibited complexes. Crystallography reveals structurally different ternary enzyme-substrate complexes that do not conform to classical mechanisms. An alternative pathway suggests substrates bind randomly, but the reaction occurs only if a specific order is followed (asymmetric cooperativity). This unreported paradigm explains substrate inhibition and reactivation by competitive inhibitors, opening new research avenues in metabolic regulation and industrial applications.

Substrate inhibition (SI) refers to a phenomenon in enzymology whereby the catalytic activity of an enzyme is reduced or is completely inhibited when the substrate concentration surpasses a certain threshold[1]. Although SI is observed in more than one fifth of biocatalysts, the underlying molecular mechanism is poorly understood[2,3]. The occurrence of SI can have significant implications for enzymatic reactions, altering the overall rate and efficiency of biochemical processes and thus the yield of biotechnological production systems[1,7]. In studies of substrate tolerance of plant enzymes, we discovered a promiscuous glycosyltransferase from *Nicotiana benthamiana* (*Nb*UGT72AY1) that exhibits an exceptionally pronounced SI for hydroxycoumarin phytoalexins (scopoletin and umbelliferone) but a moderate SI for monolignols (coniferyl aldehyde, etc.), which are precursors of lignin[8–10] (Supplementary Fig. 1C). Coumarins are a group

[1]Biotechnology of Natural Products, School of Life Sciences, Technical University of Munich, 85354 Freising, Germany. [2]KAUST Center of Excellence for Smart Health, Biological and Environmental Science and Engineering Division, King Abdullah University of Science and Technology (KAUST), Thuwal 23955-6900, Kingdom of Saudi Arabia. [3]Department of Biological Engineering, Massachusetts Institute of Technology, Cambridge, MA 02139, USA. [4]Center for Synthetic Microbiology (SYNMIKRO), Philipps-University Marburg, 35043 Marburg, Germany. [5]Department of Chemistry, Philipps-University Marburg, 35043 Marburg, Germany. [6]Leibniz Institute for Food Systems Biology at the Technical University of Munich, 85354 Freising, Germany. [7]Chemoinformatics and Protein Modelling, School of Life Sciences, Technical University of Munich, 85354 Freising, Germany. [8]Analysis and Mathematical Biology, Technical University of Munich, School of Computation, Information and Technology, 85748 Garching, Germany. [9]State Key Laboratory of Tea Plant Biology and Utilization, International Joint Laboratory on Tea Chemistry and Health Effects, Anhui Agricultural University, 230036 Hefei, Anhui, China. [10]Center for Protein Assemblies (CPA), Technical University of Munich, 85748 Garching, Germany. [11]These authors contributed equally: Jieren Liao, Umar F. Shahul Hameed. ✉e-mail: wilfried.schwab@tum.de

of important natural products that exhibit antimicrobial and anti-oxidant activity and are produced in plants as a defence mechanism against pathogen attack and abiotic stress[11]. Coumarins are valued in the pharmaceutical industry for their diverse therapeutic activities and are an active source for drug development. Glycosyltransferases are a large protein family found in all organisms, and are especially abundant in plants[12,13]. These enzymes are bi-substrate biocatalysts, i.e. they require two substrates for the reaction, as they catalyse the transfer of a sugar molecule from a carbohydrate donor to an acceptor forming *O*-, *N*-, *S*-, and *C*-glycosides and sugar esters[14].

Based on the order of binding of substrates and release of products, the reaction mechanisms of enzymatic bi-substrate reactions are classified as ping-pong and sequential mechanisms[15]. Sequential processes form a ternary intermediate complex after the two substrates have been bound in a specific order (ordered sequential mechanism) or randomly (random sequential mechanism)[16]. In plants, uridine diphosphate (UDP) sugar-dependent glycosyltransferases (UGT) are predominant. UGTs are involved in the glycosylation of proteins, lipids, sugars and of plant specific metabolites[17]. After formation of the glycosidic bond, plant UGT invert the anomeric configuration at *C1* of the sugar residue and adopt the GT-B fold, one of five folds known for UGT[18]. The sugar-donor binding site is highly conserved, facilitating the identification of the corresponding genes in different genomes[19,20]. As with many other bi-substrate enzymes, reaction rate curves deviating from Michaelis-Menten (MM) kinetics have been determined very frequently for UGT, including sigmoid substrate saturation curves as well as SI and product inhibition[2,21]. These so-called allosteric enzymes are thought to possess multiple substrate binding sites and/or multiple subunits, and can occur in different catalytically active and thermodynamically stable conformations, thus exhibiting cooperativity[22,23]. Consequently, they are controllable biocatalysts whose enzymatic activity can be modified by effectors, which is why both consumer product manufacturers and drug developers have a strong interest in such proteins.

The unusually strong SI of the recently identified uridine diphosphate-glucose (UDPG) dependent glycosyltransferase *Nb*UGT2AY1 was initially explained according to classical assumptions and confirmatory *in-silico* analyses as well as results obtained by hydrogen-deuterium exchange mass spectrometry (HDX-MS) with a second binding site for the acceptor substrate, the phytoalexin scopoletin[9]. Both an active site mutant (in which an essential amino acid in the active site was exchanged - Phe87Ile) and a chimeric enzyme exhibited reduced SI. In the chimeric enzyme, a 53-amino acid sequence had been replaced with the corresponding part from a non-inhibited homologous protein from *Solanum tuberosum*. Homology modelling of *Nb*UGT72AY1 predicted an open and closed conformation of the enzyme[9]. In a follow-up study, *Nb*UGT72AY1 was found to have UDPG glucohydrolase activity, i.e. the enzyme can transfer glucose to water in the absence of an acceptor molecule, which has also been shown for other transferases[24] (Supplementary Fig. 1A). However, this inherent catalytic side activity could be efficiently mitigated by apocarotenoids, including, α- and β-ionol and α- and β-ionone[24] (Supplementary Fig. 1D). Apocarotenoids are plant metabolites formed by enzymatic or chemical oxidative degradation of carotenoids and thus play an important role in plant defence, where they are formed together with scopoletin[25–27].

Surprisingly, the presence of apocarotenoids reduced the SI of the acceptor scopoletin, leading to increased glucoside formation[24].

Here, we show that the biogenic apocarotenoid precursor β-carotene almost completely abolish the SI of *Nb*UGT72AY1 by the acceptor, leading to reversal of the SI, whereas hydroxylated carotenoids (xanthophylls) act as inhibitors. This peculiarity now allows us to obtain crystallographic snapshots of the catalytic centre of *Nb*UGT72AY1 in the substrate-inhibited and productive states by combining the protein with different ligands. The previously known

models for sequential bi-substrate reactions, as proposed by Cleland, cannot explain these results[15]. Combining HDX-MS experiments, mathematical calculations, molecular dynamics (MD) simulations and biochemical studies using mutants, we develop an asymmetric cooperativity model that comprehensively explains the SI of the enzyme, its UDPG glucohydrolase activity and the activity-enhancing effect of the ligands at the molecular level. The information presented herein is a fundamentally important contribution to the understanding of the dynamics and molecular mechanism of catalysis of GT-B fold UGT enzymes, in particular their SI.

## Results

### Carotenoids and xanthophylls are effectors of *Nicotiana benthamiana* glucosyltransferase *Nb*UGT72AY1

*Nb*UGT72AY1 is a promiscuous transferase that glycosylates bioactive hydroxycoumarins such as scopoletin and umbelliferone, as well as monolignols involved in lignin formation and various other small molecules[8–10,24,28]. Since carotenoid-derived C13 apocarotenoids attenuate the strong SI of the enzyme by hydroxycoumarins, additional related structures were tested as likely effectors for the enzyme (Fig. 1). Chain elongation produces retinol, β-apo-8′-carotenal, and β-carotene, which in this order increasingly dampened the SI of the enzyme by scopoletin, as evidenced by the increased reaction rates after addition of the effectors (Fig. 1). The effect was concentration-dependent for retinol, whereas β-apo-8′-carotenal and β-carotene showed maximal SI suppression at 100 and 20 μM effector concentrations, respectively (Fig. 1A). The activating effect is most pronounced with β-carotene and is effective even at extremely high concentrations (1 mM) of the acceptor substrate (Fig. 1B). There is some variability in kinetics between different enzyme preparations, probably due to incorrect quantification of the enzyme. Even acyclic lycopene shows a concentration-dependent reactivation of the substrate-inhibited enzyme. In contrast, hydroxylated carotenoids, such as the xanthophylls zeaxanthin and lutein, act as inhibitors already at 20 μM (Fig. 1B). When the scopoletin concentration was kept constant at 200 μM, concentration-dependent reactivation of the substrate-inhibited enzyme by β-carotene, β-apo-8′-carotenal, and retinol was detected with maxima at 60, 40 and 200 μM of the effectors, respectively (Fig. 1C). In a previous report on *Nb*UGT72AY1, the effector-mediated reduction in SI was accompanied by a strong inhibition of the enzyme's inherent UDPG glucohydrolase activity by the effectors[24]. The newly identified effectors also inhibit the UDPG glucohydrolase activity of *Nb*UGT72AY1 in a concentration-dependent manner, with β-carotene being the most potent in suppressing glucohydrolase activity in accordance with its strong inhibition of SI (Fig. 1D). The unprecedented proof of reactivation of a substrate-inhibited enzyme by an effector provided the unique opportunity to scrutinize the mechanism of SI at the molecular level via X-ray crystallography.

### Structural analysis of apo, enzyme-substrate, enzyme-effector, and enzyme-substrate-effector complexes

To gain insight into the mechanism of *Nb*UGT72AY1 catalysis, SI and UDPG glucohydrolase activity, we determined the crystal structure of the monomeric protein (apo; Supplementary Fig. 2) and complexes revealing the structural alterations induced by different ligands such as scopoletin (acceptor), UDP (product), UDP-2-deoxy-2-fluoro-D-glucose (UDP2FG, donor substitute), β-carotene (effector), retinol (effector) and combinations thereof (Supplementary Data 1, 2). In total, nine distinct three-dimensional (3D)-structures of *Nb*UGT72AY1/ligand complexes were determined (Fig. 2).

In addition to the apo-form (I), a protein structure was obtained, which crystallized in β-carotene-containing solution (II), differed from the apo-form, and lacked the electron density for the ligand. Moreover, *Nb*UGT72AY1 crystallized in the presence of scopoletin (III), UDP (IV), and a mixture of scopoletin and the non-transferable donor UDP2FG

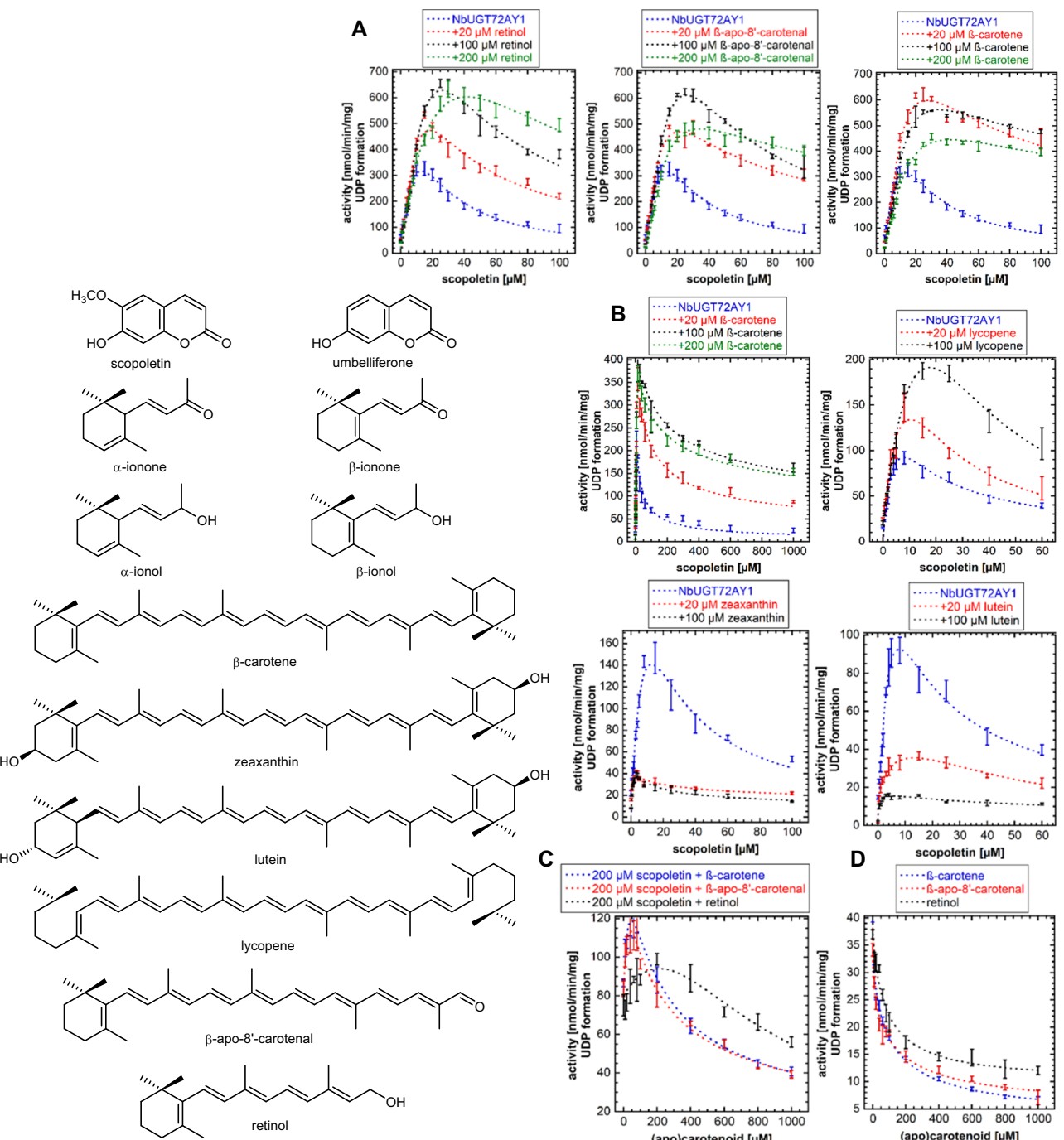

**Fig. 1 | Acceptor substrates and effectors of *Nb*UGT72AY1 and impact of effectors on the scopoletin:UDP-glucose glucosyltransferase and UDP-glucose glucohydrolase activity of *Nb*UGT72AY1. A** Different concentrations of β-carotene, β-apo-8′-carotenal, and retinol were added to enzyme assays consisting of 0-100 μM scopoletin, 100 μM UDPG and 0.5 μg *Nb*UGT72AY1 to determine the effect on UGT activity. **B** Different concentrations of β-carotene, lycopene, zeaxanthin, and lutein were incubated with increasing concentrations of scopoletin, 100 μM UDPG and 0.5 μg *Nb*UGT72AY1 to determine the effect on UGT activity. **C** Increasing concentrations of β-carotene, β-apo-8′-carotenal, and retinol were incubated with 200 μM scopoletin, 100 μM UDPG and 0.5 μg *Nb*UGT72AY1 to determine the effect on UGT activity. **D** Increasing concentrations of β-carotene, β-apo-8′-carotenal, and retinol were incubated with 100 μM UDPG and 0.5 μg *Nb*UGT72AY1 to determine their effects on UDPG glucohydrolase activity. In all cases, liberation of UDP was quantified by UDP-Glo™ assay after 30 min. Five replicates were analysed. Data are presented as mean values +/- SD. Source data are provided as a Source Data file.

(V). The protein structures and ligand binding sites were elucidated by X-ray structural analysis using molecular replacement with the crystal structure of *C*-glycosyltransferase from *Trollius chinensis* in complex with UDP (PDB:6JTD) to obtain initial phases[29]. Those structures for which data below 3 Å were obtained were subjected to refinement specific for low resolution structures as described in the methods

(Supplementary Data 1, 2). Seven different structural classes were obtained when *Nb*UGT72AY1 was co-crystallized with scopoletin, UDP2FG and retinol. They were classified into four groups based on the ligands that could be resolved (VI resolving UDP2FG & scopoletin and VII resolving just UDP2FG), orientation of the acceptor substrate (VIII), and dynamics of the major amino acid Trp350 (IX). Retinol was not

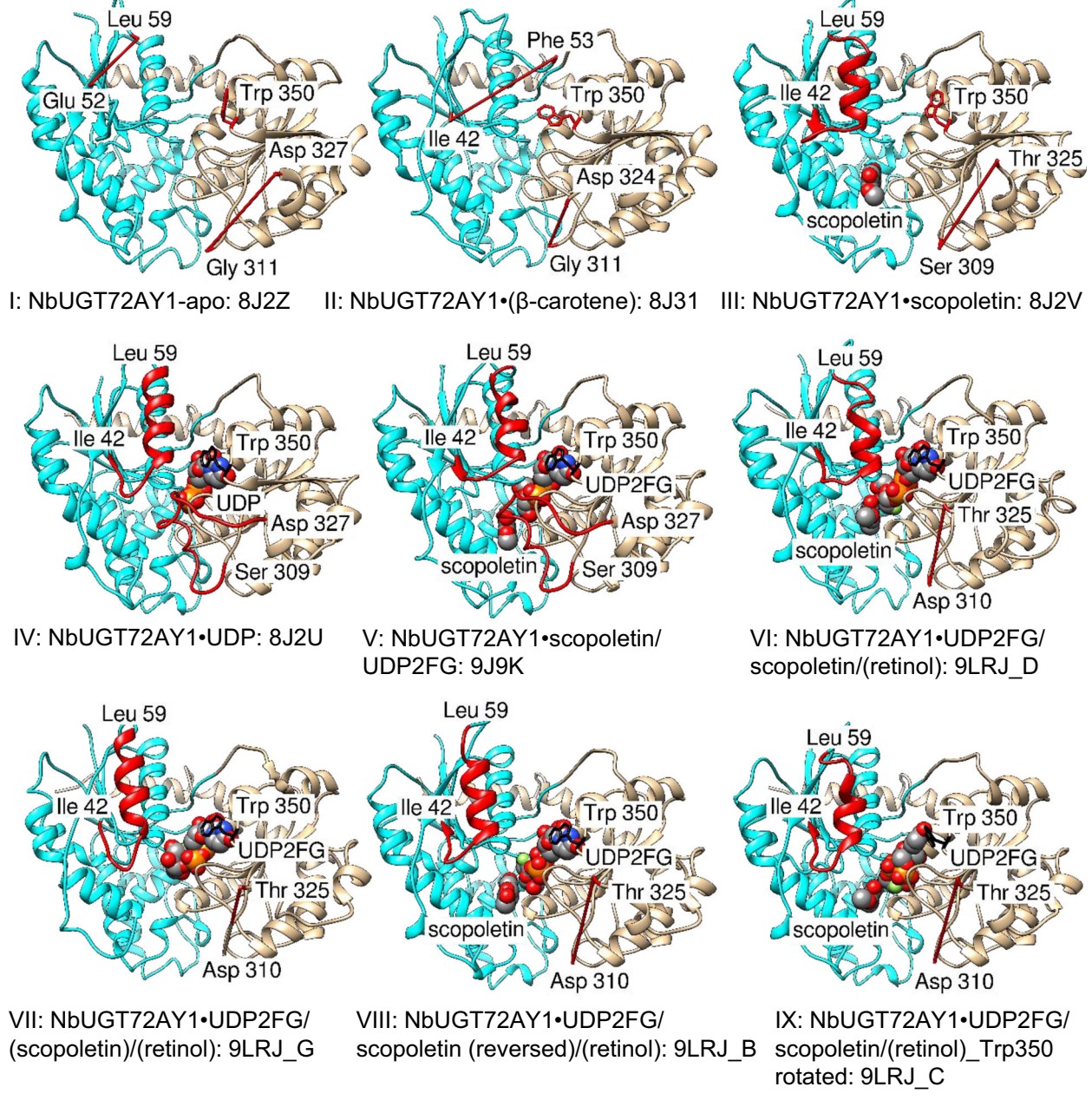

**Fig. 2 | Three-dimensional structures of *Nb*UGT72AY1 and its ligand complexes.** The N- and C-terminal domains of the UGT are depicted in cyan and brown, respectively. The acceptor substrate scopoletin, the surrogate substrate UDP2FG and the product UDP are shown as spheres, important structural features are highlighted in red, e.g. the loop/helix transition comprising Ile42-Leu59 and the closing loop at Ser309 to Asp327. Trp350, the first amino acid of the conserved PSPG motif in plant UGTs is highlighted in red and black. Red sticks represent amino acids that have not been resolved. The pdb IDs are displayed below the structures, followed by the letter of the substructure. Ligands shown in parenthesis were not visible in the X-ray crystal structure (e.g. β-carotene and retinol). Source data are provided as a Source Data file.

resolved in any crystal structure (VI to IX). Root mean square deviation (RMSD) values ranged from 0.435 to 0.824 and 0.520 to 2.206 for pruned and across all atom pairs, respectively (Supplementary Data 3).

The structural diversity of the protein in the different crystal structures was analysed by comparing the distance between Cα of all structures to the fully solved complex V (*Nb*UGT72AY1•scopoletin/ UDP2FG: 9J9K) (Supplementary Fig. 3). The most highly variable region is the α-helix, which comprises residues 42-61 (distance up to 12 Å). Five distinct GT protein folds have been identified that are capable of GT activity (termed GT-A to GT-E)[30]. *Nb*UGT72AY1 adopts a GT-B fold comprised of two distinct N-terminal and C-terminal Rossmann-like domains (shown in cyan and brown in Fig. 2, respectively) of seven and

six parallel β-sheets, respectively linked to α-helices, connected by a linker region and an interdomain cleft (Fig. 3 and Supplementary Fig. 4). The conserved plant secondary product glycosyltransferase motif (PSPG) involved in donor substrate binding is located in the C-terminus at position Trp350 to Gln393, and the catalytically active His18 together with the activating Asp118 is located in the N-terminus (Fig. 3).

**Substrate- and product-inhibited complexes share common characteristics**

In the binary complex IV (*Nb*UGT72AY1•UDP: 8J2U) and the ternary complex V (*Nb*UGT72AY1•scopoletin/UDP2FG: 9J9K) most of the

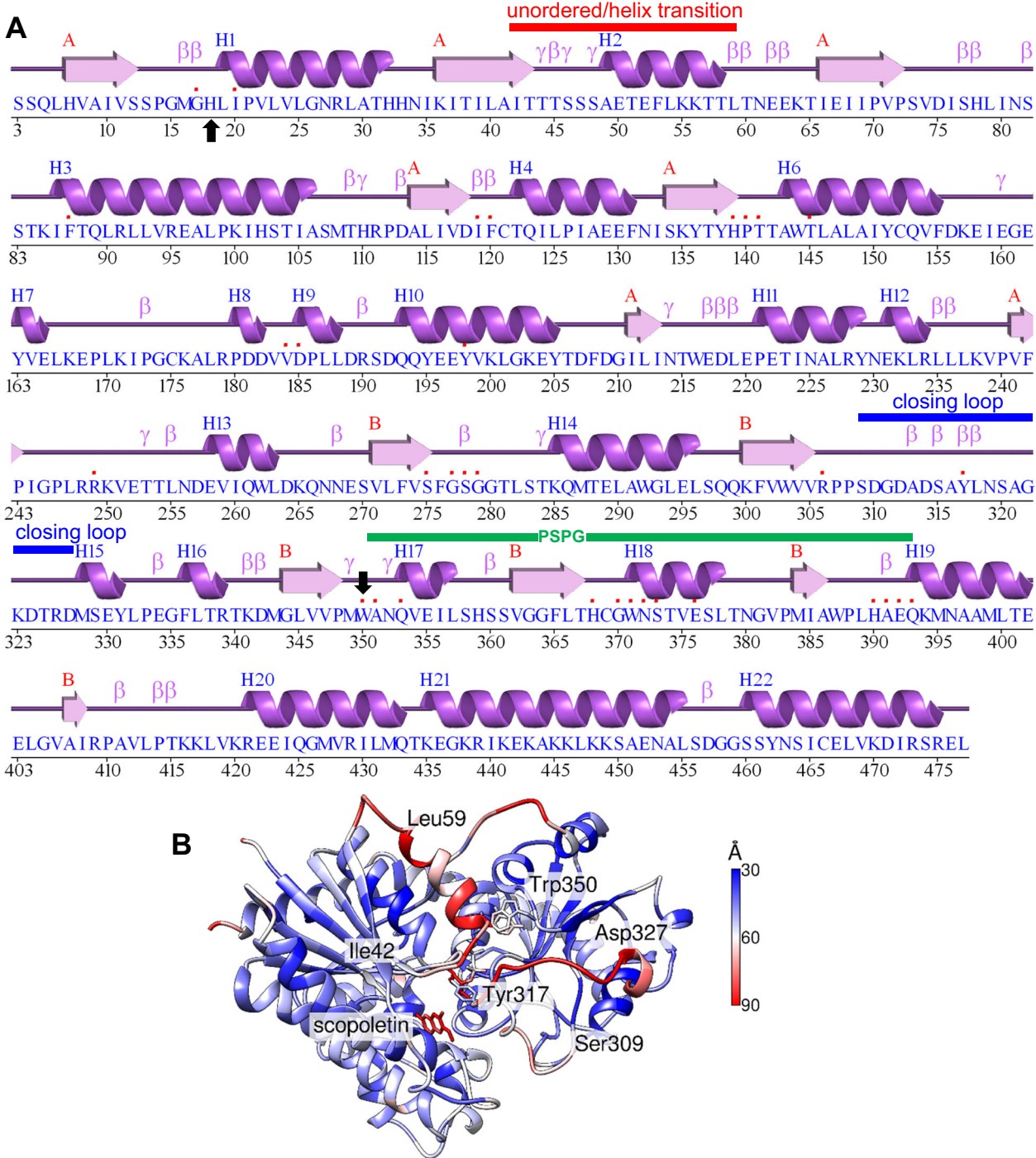

**Fig. 3 | Secondary structure of *Nb*UGT72AY1 based on complex V: *Nb*UGT72AY1•scopoletin/UDP2FG: 9J9K and the B-factor representation. A** Helices are numbered H in ascending order and sheets of the N- and C-terminal domains are numbered A and B, respectively. The β and γ designations stand for beta and gamma turns, respectively and contacts of residues to ligands are indicated by a small red box. Generated by PDBsum (https://www.ebi.ac.uk/thornton-srv/ databases/pdbsum/Generate.html). A green, red and blue bar indicates the plant secondary product glycosyltransferase (PSPG) motif, the unordered/helix transition region and the closing loop segment, respectively. A black arrow marks catalytically active His18 and Trp350. **B** Representation of the B-factors (in Å²) as specified in PDB 9J9K.

protein residues are visible in the electron density maps. Conversely, in all other complexes at least one sequence segment is absent, suggesting substantial mobility or flexibility (red sticks in Fig. 2). Complex V and IV represent the SI and product-inhibited enzyme complexes, respectively, as they were obtained in the presence of an excess of substrate (scopoletin) and product (UDP), respectively (Supplementary Fig. 5). Asn372, Ser373, Glu376, Arg249, Ala351, and Trp350 frame

their donor binding sites and in both structures, the indole ring of Trp350 points in the direction towards the uracil structure of UDP2FG and UDP, resulting in a π-stacking interaction (Supplementary Fig. 5, 6). All other complexes that vary in their bound ligands have one (III, VI, VII, VIII, and IX) or two (I and II) disordered sequence parts (shown as red sticks in Fig. 2) located between Ile42-Leu59 and Ser309-Asp327 (Fig. 3A). These two sequence segments and the dynamics of Trp350

(the first amino acid of the PSPG box) differ significantly in the structures obtained; illustrating the dynamic changes that take place during catalysis. Consistent with published UGT structures for apo-enzymes showing 40% sequence similarity (PDB: 2O6L, 6JEL, 6SU7, and 7ERY) and complexes with acceptor molecules (PDB: 5NLM, 7CJX, and 7YAN), the benzene ring of Trp350 also points outward, away from the active site, in the apo form of *Nb*UGT72AY1 and its complex with scopoletin (Fig. 2, I and III). However, when UDP or UDP2FG is bound, Trp350 forms a π interaction with the uracil ring system of the ligand (Fig. 2, IV-VIII, except IX).

### Ligand binding alters the conformation of the disordered regions

The sequence of Ser309-Asp327 represents the active site loop known from other enzymes[31] and UGT[22,32], which closes the catalytic centre and removes interfering water prior to reaction. The Ile42-Leu59 segment is not resolved in the 3D structure of the apo-enzyme (I: 8J2Z) and crystals grown in the presence of β-carotene (II: 8J31). However, Ile42-Leu59 adopt an α-helical conformation (H2 in Fig. 3, Supplementary Fig. 7) when at least one of the two substrates is present (III-IX). Thus, helix H2 formation is clearly associated with catalysis. Upon substrate binding, the disordered Ile42-Leu59 sequence part in the apo-protein transforms into helix H2, which aligns parallel to helix H1 above the catalytic site. This structural alteration explains a result from a previous hydrogen/deuterium mass spectrometry (HDX-MS) analysis that showed that helix H1 incorporates less deuterium when scopoletin and/or UDP are bound to the apo-enzyme (Supplementary Fig. 8). Hence, the transition to H2 protects helix H1 from isotope exchange and makes H1 less flexible[9]. Helix H1 carries the catalytically active His18 (Supplementary Fig. 7A), thus, the transition to H2 directly affects reaction kinetics. However, since the addition of apocarotenoids increased the H/D exchange in H1[24], these effectors reverse the helix-forming effect of scopoletin and the corresponding H2 amino acids should therefore remain flexible. Consistent with this hypothesis, electron density maps derived from crystals grown with β-carotene failed to resolve the secondary structure between Ile42 and Phe53 (Fig. 2; complex II). A conformational change in the secondary structure upon addition of ligands was also detected by circular dichroism (CD) spectroscopy (Supplementary Fig. 9). The CD spectra show that the addition of β-carotene to the protein reduces the estimated proportion of α-helices from 29% to 17%, while the content of turns increases to a comparable extent (9% to 18%), i.e. the carotenoid increases the flexibility of the protein. In contrast, the addition of scopoletin promotes the formation of α-helices (29% to 31%) and β-sheets (28% to 31%), confirming the structural differences between the apo-protein (I) and the binary NbUGT72AY1•scopoletin complex (III), as determined by crystallography. In contrast, the simultaneous addition of β-carotene considerably reduces the proportion of α-helices (31% to 23%) and β-sheets (31% to 23%). It appears that β-carotene keeps the protein in a dynamic state and acts as a 'molecular plasticizer'.

### Tyr317 likely plays a role in the catalytic cycle

The catalytic mechanism of inverting UGT such as *Nb*UGT72AY1 is a direct displacement bimolecular nucleophilic substitution (SN₂)-like reaction enabled by an enzymatic base catalyst[33]. His18 is 4.0 Å apart from *O*7 of scopoletin in complex V and presumably plays the role of a catalytic base for the deprotonation of the acceptor to allow the nucleophilic attack at the anomeric centre of the donor (Supplementary Fig. 10A, B, E). In many GT-B fold enzymes, protonation of the His is subsequently stabilized by interaction with an Asp. In *Nb*UGT72AY1, Asp118 is located at a distance of 4.8 Å, while Tyr317, an amino acid of the closing loop, is only 2.8 Å apart (Supplementary Fig. 10A). Thus, Tyr317 could contribute to the activation of catalytic His18 with the participation of the β-phosphate group of UDP-glucose

(Supplementary Fig. 10C). A similar mechanism involving the α-phosphate of the sugar donor in the catalysis of a human *O*-linked β-*N*-acetylglucosamine transferase has been proposed[34]. Since the Tyr317Phe mutant showed higher catalytic activity than the wild type enzyme (Supplementary Data 4), this mechanism seems unlikely. However, due to its close proximity to N7 of His18 in the Michaelis complex (3.9 Å), Tyr317 could also serve as an alternative sugar acceptor, which would explain the increased activity of the Tyr317Phe mutant (Supplementary Fig. 10D). Glucose *O*-glycosidically bound to tyrosine has already been detected in UGT74F2 from *Arabidopsis thaliana*[35]. However, it could also be a posttranslational modification catalysed by protein glycosyltransferases.

### Structures obtained with retinol in the crystallization solution have special features

Because addition of retinol to enzyme assays considerably reduces the SI of scopoletin in *Nb*UGT72AY1 (Fig. 1), crystals grown in (apo)carotenoid-containing solution are expected to represent productive protein complexes (Supplementary Fig. 5). Co-crystallization of *Nb*UGT72AY1 with scopoletin, UDP2FG and retinol yielded seven different structures (A-G), which were grouped into four classes (VI – IX) according to their molecular similarities (VI_A, VI_D, VI_F, VII_E, VII_G, VIII_B, and IX_C; Supplementary Fig. 11, 12). Although retinol and the closing loop between Ser309 and Thr325 were not visible in all structures, the spatial arrangement of helix H2 (Ile42-Leu59), the direction of rotation of Trp350, the orientation of the acceptor substrate, and the number of ligands differed among the four groups (Fig. 2). Thus, complex VI (VI_A, VI_D, and VI_F; Supplementary Fig. 11, 12) presumably constitutes the productive 'Michaelis' structure, while VII (VII_E and VII_G, Supplementary Fig. 11, 12) represents the precursor complex, which is still lacking the acceptor substrate. Since in VIII (VIII_B; Supplementary Fig. 11, 12) the scopoletin is rotated by 180° in the horizontal axis and thus the distance to His18 increases to 8.3 Å, and in IX (IX_C; Supplementary Fig. 11, 12) the Trp350 adopts a spatial arrangement that differs from all other structures, it is reasonable to assume that both VIII and IX are non-productive structures.

### Crystals formed in the presence of (apo)carotenoids show common properties

In all *Nb*UGT72AY1 crystals obtained in solutions with retinol or β-carotene, the amino acid sequence between Ser309 and Asp327 (closing loop) is disordered regardless of whether acceptor and/or donor substrates are present (Fig. 2; II, VI-IX). Consequently, in productive enzyme complexes, this region is extremely flexible, but in SI (V) and product-inhibited structures (IV), much of the enzyme population appears to be in the closed form (Supplementary Fig. 13). This observation is confirmed by results on a substrate-inhibited adenylate kinase (AK), which catalyses the phosphotransfer from ATP to AMP[4]. Single-molecule Förster fluorescence resonance energy transfer (FRET) spectroscopy results revealed that inhibitory AMP concentrations lead to faster and more cooperative domain closure by ATP, resulting in an increased population of the closed state of AK. As will also be shown later for *Nb*UGT72AY1, the effect of acceptor binding on AK could be modulated by mutations, whereas the SI effect in *Nb*UGT72AY1 could equally be influenced by mutations and effectors. Although all the crystals that formed in the presence of 0.5 mM β-carotene and retinol were clearly different from their (apo)carotenoid-free counterparts (I and V versus II and VI, respectively), the effector molecules were not detectable in the electron density map and are invisible in the 3D structure (Fig. 2). This may indicate that the effectors bind in a fuzzy way, possibly through poorly directional hydrophobic interactions, leading to multiple positions and orientations in the crystal that are equally likely, or that the effector occupancy of protein molecules in the crystal is too low. Therefore, molecular docking was performed to predict the interaction domains of the protein with the

β-carotene and retinol effector (Supplementary Fig. 14A, 15A, respectively). In both cases (β-carotene and retinol docked on II and VII, respectively), a majority of the ligand population occupies the hydrophobic acceptor-binding site in the cleft representing the active site, while a smaller number is bound to nonpolar wells on the surface of the protein. The electron density between His18, Phe87 and His139 is low in these structures compared to other structures, so that these residues become more dynamic due to the (apo)carotenoid binding. The planes of the cyclohexene rings of the (apo)carotenoids, including β-apo-8'-carotenal, and the aromatic system of the hydroxycoumarin are in each case congruent, which explains the coincident effects of the effectors with this structural element (Supplementary Fig. 16, 17). Xanthophylls such as lutein and zeaxanthin carry hydroxyl groups in position 3 and 3'. These could interact with His139, Thr141 and Thr145 via polar interactions, as they are located at a distance of 3.1–4.1 Å to carbon 3 in the xanthophylls (Supplementary Fig. 16E), which could eventually lead to the strong inhibition of the enzyme observed in Fig. 1B.

## Comparison of productive and non-productive complexes reveals major differences

Examination of the *Nb*UGT72AY1 crystals obtained in the absence and presence of different ligands revealed three main features in which the structures differ (Supplementary Fig. 18A). Firstly, there is helix H2, which only forms when the acceptor or donor substrate is bound, secondly the loop, which closes the active site in the SI and product-inhibited complexes, and thirdly the orientation of the first amino acid of the PSPG box, which only establishes a π interaction when UDP or UDP2FG binds. The product inhibited protein structure IV and SI complex V show closed active sites with distinct electron densities for the residues from 310-326 in IV and V, while catalytically active VI and VII lack electron densities for this region due to the high dynamics and the loop may remain open (Fig. 2; Supplementary Fig. 13, 18D). The shift in population sizes toward closed forms for the SI protein has already been shown for adenylate kinase using FRET analyses[4]. The comparison of the crystal structures of *Nb*UGT72AY1 also shows that the dynamics/mobility of the first amino acid of the PSPG box is likely to play a role in catalysis, albeit not a major one, as later biochemical analyses of a Trp350Ala mutant will show. In all UDP/UDP2FG-bound structures (IV, V, VI, and VII), the indole ring system of Trp350 forms a π interaction with the uracil ring. The arrangement of the Trp350 side chain in the 3D structure obtained in the presence of scopoletin (III) is clearly different (Supplementary Fig. 18B). Ligand binding influences the length of helix H2. In all structures that have scopoletin bound (III, V, VI, VIII, and IX), H2 has 2-3 turns while conformers associated with UDP (IV) and UDP2FG (VII) have 4 turns (Fig. 2). In addition, the RING server (https://ring.biocomputingup.it/) was used to calculate non-covalent molecular interactions based on geometric parameters in the *Nb*UGT72AY1 structures for comparison[36]. The results show that the *Nb*UGT72AY1 complexes can be clearly distinguished from each other according to their bound ligands by the pattern of non-covalent interactions determined based on their crystal structures (Supplementary Fig. 19). The lowest number of interactions (total edges) was calculated for the protein in the presence of β-carotene (**II**), while the highest number of bonds was obtained after addition of UDP (IV_A and IV_B), which is consistent with the assumption that the carotenoid promotes protein dynamics, while in the product-inhibited complex the mobility of the protein is hindered. The complexes with UDP2FG (VII_E and VII_B) show significantly more interactions within the protein than the structures with scopoletin (III), which is due to the increased number of van der Waals interactions. On the other hand, the SI complexes (V_A and V_B) exhibit more H-, π-π stacking and π-cation bonds, but less van der Waals interactions compared to the catalytically active complexes VI_A, VI_D, and VI_F), leading to similar overall interactions. From this we conclude that both the binding of

scopoletin and UDP2FG lead to conformational changes of the protein but yield different complexes. No more than one scopoletin molecule could be detected in any of the crystal structures obtained, which makes the hypothesis of a second binding site of the acceptor molecule as the cause of SI unlikely[9]. Isothermal titration calorimetry (ITC) measurements confirmed a 1:1 stoichiometry of protein and scopoletin and yielded a $K_d$ of $4.5 \pm 1.1\,\mu M$ (Supplementary Fig. 20) and molecular dynamics (MD) simulations demonstrated the flexibility of Trp350, with UDPG fixing the spatial orientation of Trp350 (Supplementary Data 5; Supplementary Note 1). Hydrogen/deuterium exchange mass spectrometry (HDX) showed that scopoletin and (apo)carotenoids bind at similar positions in the protein (Supplementary Note 2) and melting curves indicated different thermostability of the protein in combination with substrates and ligands (Supplementary Note 3).

## Snapshots of catalysis allow deriving a revised mechanism of SI

From the previous discussion, we derive a hypothesis for the SI mechanism of the bi-substrate enzyme *Nb*UGT72AY1. For this purpose, the snapshots of the catalytic steps as shown in X-ray crystal structures I-IX (Fig. 2) were combined to form a mechanistic model (Fig. 4). Starting with the apo-enzyme (I), two different conformational changes can occur after substrate binding depending on whether the acceptor (I → III) or donor (I → VII) is the first to interact with the protein (Fig. 4). Structures III and VII differ both in the length of helix H2 and in the spatial orientation of Trp350, leading to the formation of an unproductive SI complex V and a catalytically active complex VI, respectively, after binding of the respective second substrate (Fig. 2). Thus, as the crystal structures indicate, random binding of substrates to the apo-enzyme is possible, but the products are formed only after ordered sequential binding of the substrates. High scopoletin concentrations lead to increased use of the non-productive cycle, eventually resulting in SI. (Apo)carotenoids play a role as placeholders. They block the acceptor-binding site with minor conformational change of the protein (I → II) until the donor is associated with the enzyme and can then be competitively replaced by scopoletin (VII → VI).

Without addition of acceptor substrate, (apo)carotenoids act as inhibitors of the inherent UDPG glucohydrolase activity (Fig. 1D), whereas high concentrations of (apo)carotenoids (greater than 200 μM for β-carotene, β-apo-8'-carotenal, and retinol and greater than 500 μM for α- and β-ionols and α- and β-ionones) inhibit the glucosyltransferase activity of *Nb*UGT72AY1 (Fig. 1C; Supplementary Fig. 21), consistent with their function as competitive inhibitors.

## Kinetic analysis of *Nb*UGT72AY1 reveals cooperativity for both substrates

Since a second acceptor-binding site could not be confirmed, the kinetic equation was revised to explain SI in *Nb*UGT72AY1 (Supplementary Note 4). Furthermore, we performed studies at lower amounts of UDPG because *Nb*UGT72AY1 showed SI for scopoletin at elevated donor substrate concentration (100 μM) (Fig. 5A).

Equation 2 (Supplementary Note 13) explained the values obtained very well, with both the $K_S$ value for scopoletin (19–21 μM) and the Hill coefficients ($n$ and $x$) being independent of the amount of UDPG added (Supplementary Data 4). Only the maximum reaction rate $v_{max}$ increased with donor substrate concentration. A graph plotting the respective UDPG concentrations versus the calculated $v_{max}$ values showed a hyperbola, from which the $K_D$ value for UDPG at 19 μM could be calculated as well as the maximum possible reaction rate $v_{max} = 1157$ nmol/min/mg if no SI would occur (Fig. 5C). Furthermore, we determined the reaction rates for UDPG at defined constant scopoletin concentrations (Fig. 5B). The kinetic parameters were calculated using equation 3 (Supplementary Note 13). In this case, as already shown by the SI kinetics of scopoletin, the highest $v_{max}$ value was obtained with 25 μM acceptor substrate. Higher concentrations of 50 and 100 μM

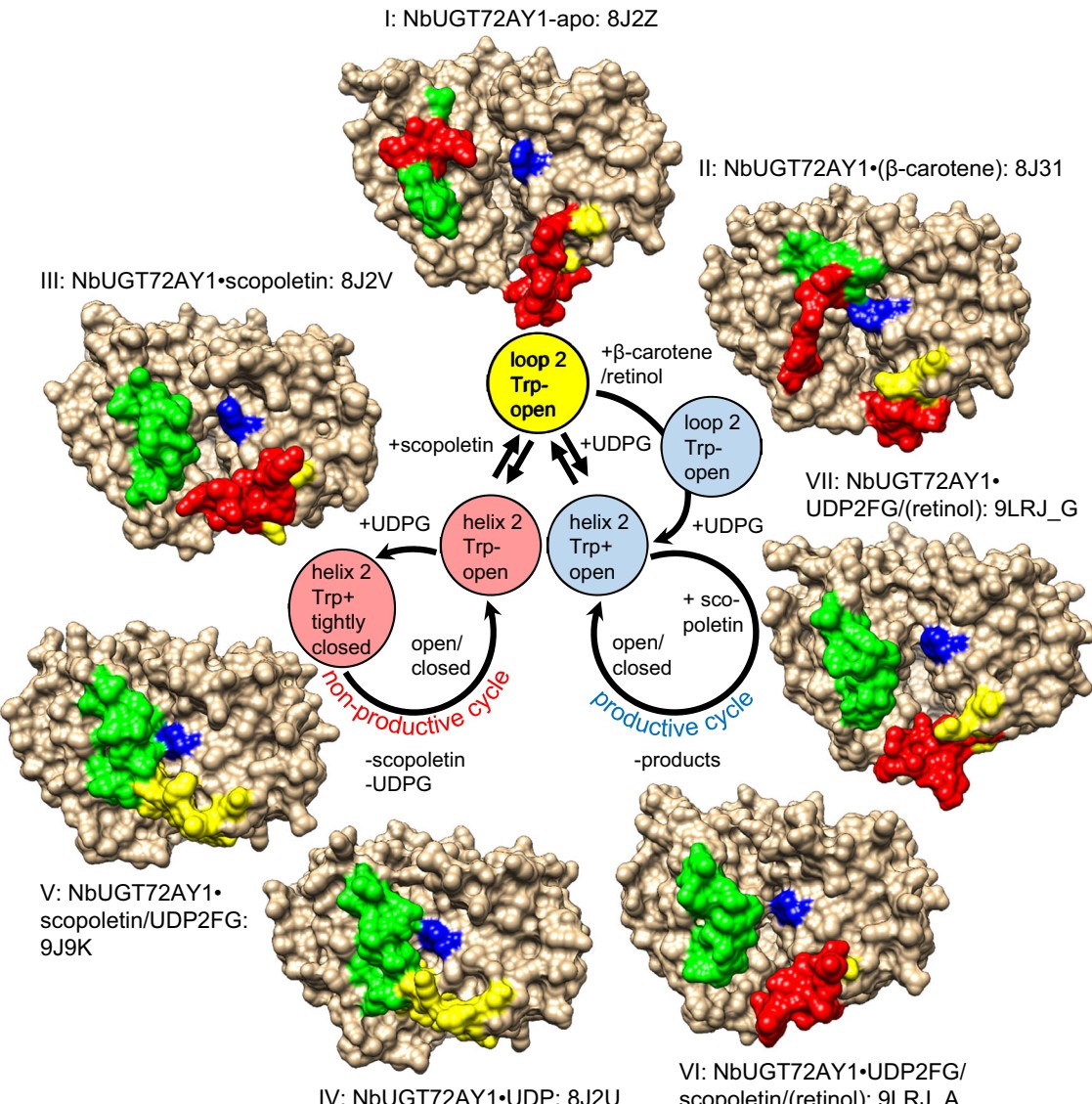

**Fig. 4 | Derivation of the proposed reaction mechanism from the identified enzyme conformers.** In the crystals, unresolved structures that were predicted are colored in red, the region of the loop/helix transition (Ile42-Leu 59) in green, Try350 in blue, and the closing loop (Ser309-Asp327) in yellow (e.g. complex I showing the apo protein). Binding of scopoletin (III) or UDP2FG (VII) results in the conversion of the loop (loop 2) between Ile42-Leu59 (green) into a helix (helix 2), but differ in the orientation of Try350 (blue) (Trp-, π interaction with uracil is not possible; Trp + , π interaction), whereas the closing loop remains flexible in both structures (yellow/red). The non-productive complexes IV and V are compact, fully resolved, and display a closed active site. In the presence of β-carotene (II) and retinol (VI and VII), the 3D structures are more open and the loop covering the catalytic center is not fully resolved. These latter complexes are actively involved in catalysis as clearly demonstrated by the increase in enzymatic activity after apo/carotenoid addition.

scopoletin inhibited the enzyme. While the $K_U$ value was independent of the scopoletin concentration, the Hill coefficient ($h$) for UDPG increased sharply with increasing acceptor amount (Supplementary Data 4; Fig. 5D). This result can be explained by an observation made in a study of AK, where inhibitory concentrations of the acceptor substrate AMP led to faster and more cooperative domain closure by the donor substrate ATP, which in turn led to an increased population of the closed state[4]. A Hill coefficient $h = 1$ was determined at approximately 20 μM scopoletin. In this case, equation 3 corresponds to a Michaelis-Menten relationship and the maximum achievable reaction rate can be calculated. Mechanistically, at this scopoletin concentration, the apo-enzyme must have been completely converted into the two conformers (helix H2 may then have formed in all monomers) and the maximum concentration of catalytically active conformer must have been generated. If the acceptor substrate is further increased,

conversion of the catalytically active to the inactive enzyme form occurs. At $h < 1$, apo-enzyme is presumably still present, while at $h > 1$ a preferential conversion to the inactive form takes place. Scopoletin acts as a non-competitive inhibitor of UDPG in a random sequential single displacement reaction according to a Lineweaver-Burk plot (Fig. 5E).

### Interaction network analysis identifies an allosteric communication pathway

Amino acids identified by mutational analysis as important for SI in *Nb*UGT72AY1 (Supplementary Note 5, Supplementary Data 4) were highlighted in complex V (Fig. 5F), and interaction network analysis of the SI-inhibited enzyme was performed (Fig. 5G, H). A π-π interaction chain was identified linking Tyr198, Phe87 and Phe120 of the acceptor-binding site to His390 and His368 of the donor binding site and His18

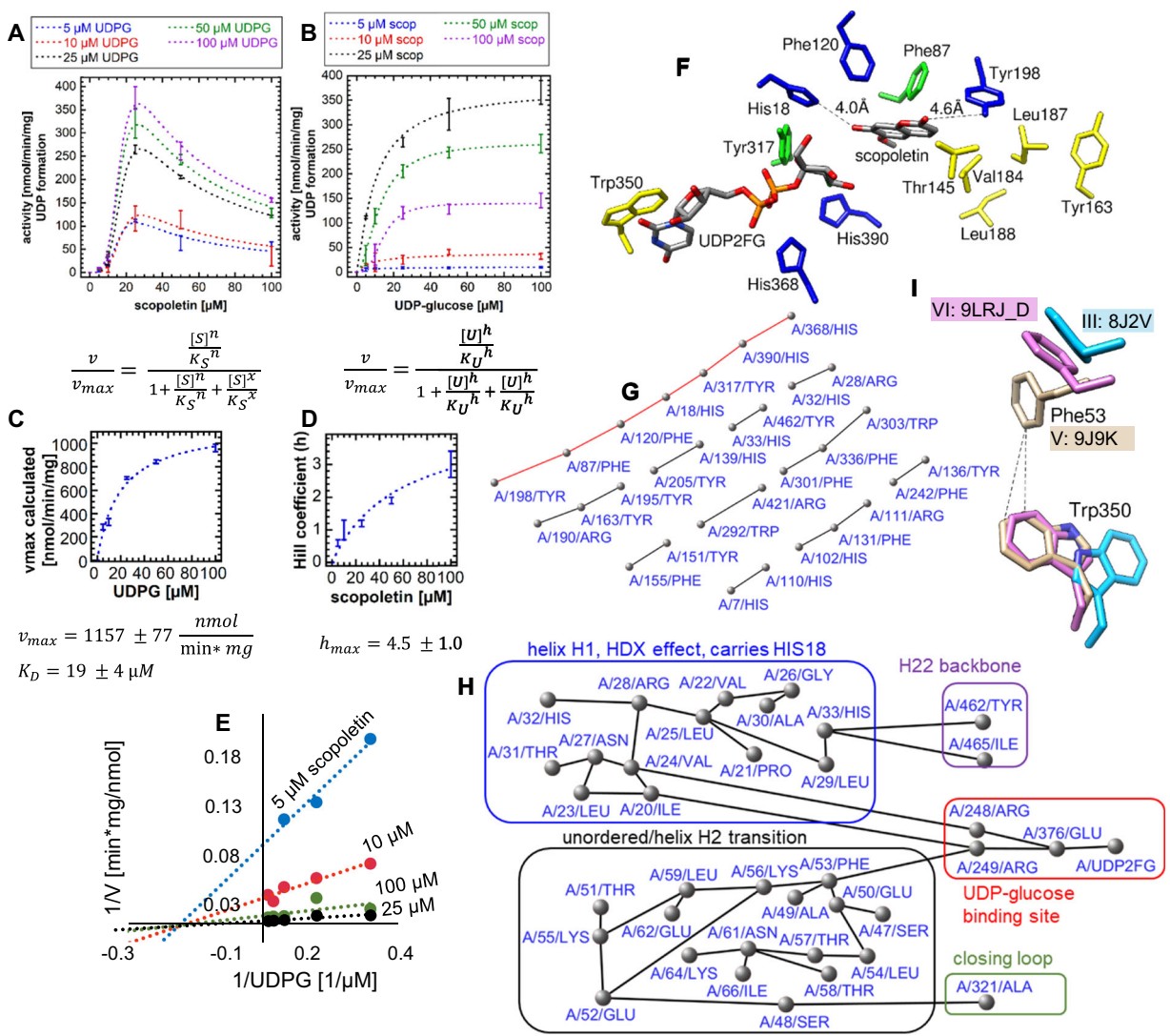

**Fig. 5 | Enzyme kinetics of the bi-substrate enzyme *Nb*UGT72AY1 and prediction of the allosteric communication pathways in *Nb*UGT72AY1.** A matrix analysis was performed in which increasing concentrations of scopoletin (5, 10, 25, 50 and 100 μM) were incubated with increasing concentrations of UDPG (5, 10, 25, 50, and 100 μM). (Supplementary Data 4). Five technical replicates of each composition were analyzed. Data are presented as mean values +/- SD. **A** Increasing concentration of acceptor substrate scopoletin at defined concentrations of donor substrate UDPG and the equation used for the calculation of the reaction rates. *n, x* Hill coefficients calculated for scopoletin. **B** Varying concentrations of UDPG at defined concentrations of scopoletin and the equation used for the calculation of the reaction rates. *h* Hill coefficient calculated for UDPG. **C** Calculation of the $K_D$ for UDPG derived from the values determined in (**A**). **D** Dependence of Hill coefficient

*h* determined in (**B**) on the scopoletin concentration. **E** Lineweaver-Burk plot. Scopoletin acts as a non-competitive inhibitor for UDPG in a random sequential single-displacement reaction. **F** Amino acids (yellow and green) in the binding pockets of scopoletin and UDP2FG shown to contribute to SI in *Nb*UGT72AY1, and amino acids (blue and green) forming a π-π interaction chain in the residue inter-action network constructed by Cytoscape. **G** Intramolecular π-π interactions in *Nb*UGT72AY1 complex V predicted by Cytoscape. The allosteric information pathway connecting the scopoletin (Tyr198) and UDP2FG binding sites (His368) is highlighted in red. **H** Interaction network calculated with Cytoscape showing the amino acids that are likely to contribute to SI. **I** Hydrophobic interaction of Phe53 and Trp350 in complex V: 9J9K. Source data are provided as a Source Data file.

and Tyr317 involved in catalysis (Fig. 5G). Val184, Leu187, Leu188 and Tyr163, the amino acids important for SI form a hydrophobic pocket in which Tyr198 is embedded. Tyr198 stabilizes the acceptor by hydrogen bonding, which may explain the effect of the above amino acids on the SI of the enzyme (Fig. 5F). The π-π interaction chain is arranged in a spiral around the longitudinal axis of the substrates, so that after completion of the chain by loop closure and thus insertion of Tyr317, a very rigid stable structure is formed. Furthermore, network analysis visualized the information pathway from helix H22, part of the back-bone and helix H1, bearing the catalytically active His18, through amino acids of the UDPG binding site (Arg248, Arg249, and Glu376), region H2, which becomes a helix upon ligand binding, to Ala321, part of the closing loop (Fig. 5H). Phe53, an amino acid of helix H2, interacts via π-cation binding with Arg214 of the UDPG-binding pocket and

approaches Trp350 4.1 Å in SI complex V (Fig. 5I) and 4.0 Å in the product-inhibited structure IV, while the distance is larger in all other structures. Aside from the closed active site, the amino acids in the nonfunctional structures move closer together, resulting in a more compact complex and a higher number of non-covalent interactions (Supplementary Fig. 19). We also established that *Nb*UGT72AY1 shows hysteresis (Supplementary Note 6).

**Transgenic β-carotene overproducing plants produce significantly higher amounts of scopolin than wild type plants after induction of scopoletin production**

After administration of scopoletin and apocarotenoids to in vitro maintained *N. benthamiana* plants, the increased production of sco-poletin glucoside (scopolin) in the presence of apocarotenoids has

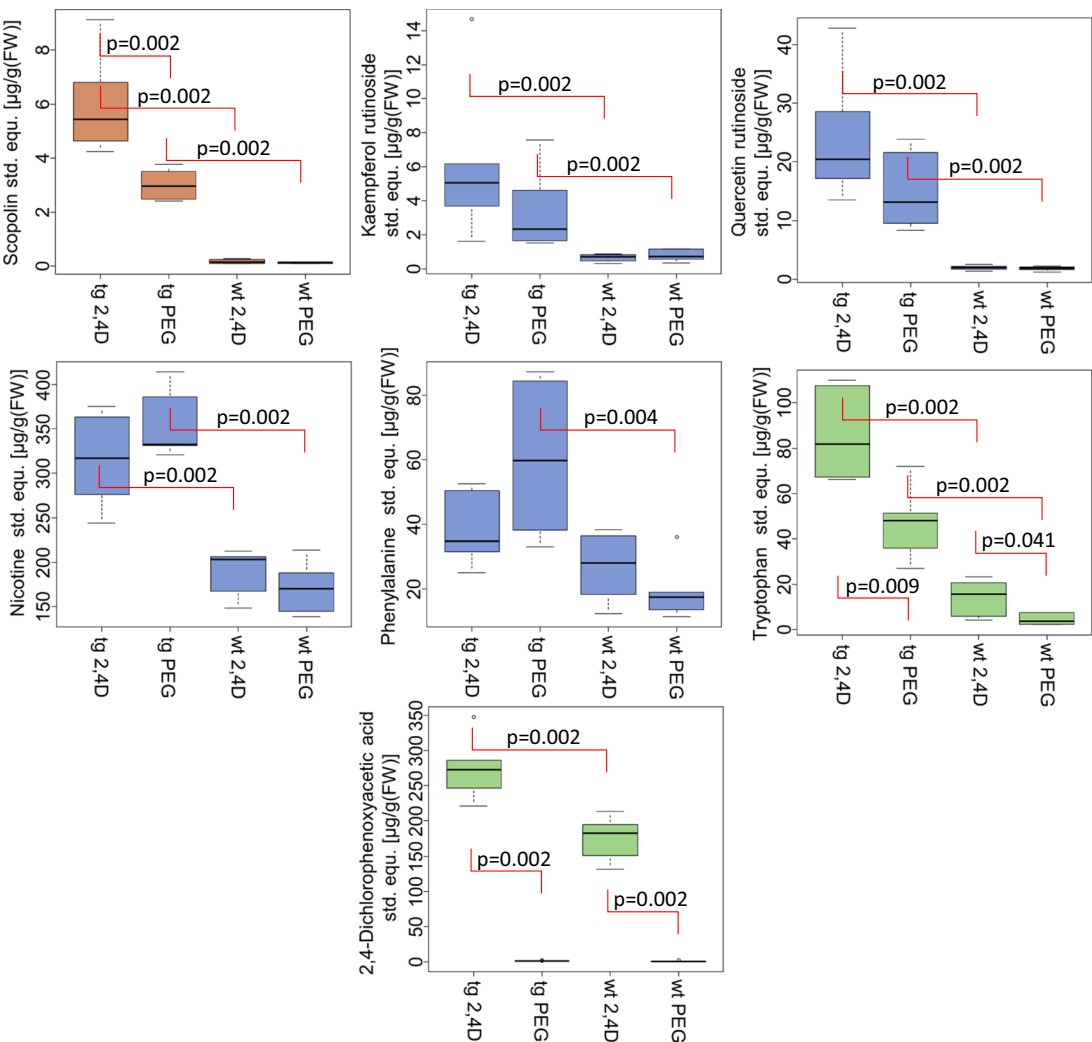

**Fig. 6 | Relative concentrations of metabolites shown as boxplots after treatment of wild type (wt) and transgenic β-carotene overproducing *N. tabacum* cv. *Xanthi* plants (tg) with aqueous solutions of polyethylene glycol (PEG) and PEG/2,4-dichlorophenoxyacetic acid (2,4D).** Plant metabolites were determined by LC-MS and relative concentrations calculated as μg-equ. of the internal standard / g fresh weight (FW). The p-values were calculated using the two-sided Wilcoxon rank sum test in R 4.2.2. Box plots present the median and 25th and 75th percentiles (whiskers: +/− 1.5*interquartile range; circles are outliers). Six biological replicates were analysed. Source data are provided as a Source Data file.

already been demonstrated[24]. This result supports the effector effect of apocarotenoids on *Nb*UGT72AY1 and indirectly shows that the SI of the enzyme in the plant might play a role. Since the administration of scopoletin via the root system is a rather artificial delivery system, we sought a model that more closely resembles the natural situation.

Using wild type and transgenic *N. tabacum* cv. Xanthi plants that exhibited increased levels of β-carotene[37,38], we induced the production of scopoletin and scopoletin glucoside by briefly immersing the leaves in a 2,4-dichlorophenoxyacetic acid (2,4D)-containing aqueous polyethylene glycol (PEG) solution[39]. Thus, without injuring the plants, scopoletin glucoside formation could be stimulated in the presence of low and high β-carotene concentrations. Although no scopolin could be detected in untreated wild type and transgenic plants, PEG and 2,4D/PEG treated wild-type plants produced detectable levels (0.2 μg-equ/g fresh weight) of the hydroxycoumarin glucoside (Fig. 6). In contrast, the transgenic, β-carotene-overproducing plants treated with PEG accumulated more than 10 times the amount of the glucoside (3.0 μg-equ/g fresh weight) and after the application of 2,4D/PEG the concentration increased again significantly (5.9 μg-equ/g fresh weight). The transgenic plants produced significantly more scopolin than the wild type plants, which was attributed to the increased β-carotene

concentration. Other glycosides such as the kaempferol and quercetin rutinoside also showed similar behaviour although the increase was not as strong and the effect of 2,4D/PEG versus PEG treatment was not significant ($p > 0.05$). Furthermore, various metabolites associated with plant defence such as nicotine, phenylalanine, and tryptophan accumulated in the transgenic plants and after treatment. Overall, the transgenic plant contained a higher content of specialized metabolites. Although the effect of β-carotene and SI of *Nb*UGT72AY1 *in planta* cannot be proven beyond doubt even with this experiment, the results do not contradict the postulated hypothesis, but rather support it.

## Discussion

Enzymes play a crucial role in biological systems as vital catalysts that facilitate and regulate biochemical reactions. Although an increase in substrate concentration generally leads to an increase in enzyme activity, cases of SI have been observed[2]. The occurrence of SI can have a significant impact on the overall rate and efficiency of biochemical processes and has been observed in more than 20% of biocatalysts. Several mechanisms have been discussed as the cause of SI, including at least two binding sites within the enzyme protein, formation of a ternary dead-end enzyme complex, and ligand-induced changes in

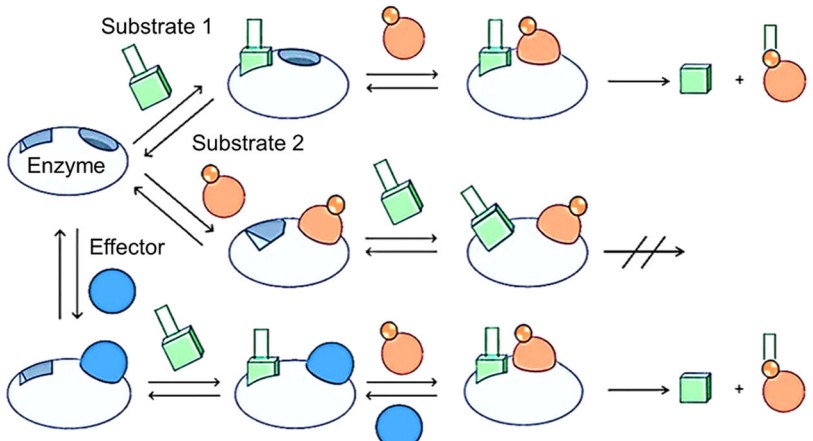

**Fig. 7 | Substrate inhibition in *Nb*UGT72AY1.** Binding of both the donor substrate 1 and the acceptor substrate 2 result in conformational changes in the enzyme. In contrast to the predictions of the Cleland model[15], however, the structural rearrangements differ depending on the order of substrate binding and lead to different ternary complexes. When donor 1 binds as the first substrate, the induced structural changes yield an active enzyme-substrate complex. A catalytically inactive enzyme-substrate complex is formed only when acceptor 2 is bound first. Alternatively, (apo)carotenoids (effector) can serve as placeholders to allow substrate 1 to bind to the protein without a conformational change previously triggered by substrate 2. Prior to catalysis, substrate 2 replaces the effector. In the random sequential model of bi-substrate binding of enzymes, the two branches leading to the ternary complex are usually considered equivalent[15]. At the same time, however, it has been shown that the binding of substrates triggers a ligand-dependent conformational rearrangement of the enzyme structure, with structural changes related to the sequential binding of substrates. Consequently, enzyme-substrate complexes with different catalytic properties result.

enzyme conformation, but none of these mechanisms applies to *Nb*UGT72AY1. Although *Nb*UGT72AY1 is a bi-substrate enzyme, the crystal structures contain only one binding site each for the acceptor and the donor substrate so that a second substrate cannot inhibit the reaction. Although a ternary dead-end complex V was identified in our study, it differs from the previously postulated dead-end complex in that UDPG and not UDP is bound. Furthermore, the observed conformational change in *Nb*UGT72AY1 does not prevent binding of the acceptor or donor substrate. There must be another explanation for the occurrence of SI in *Nb*UGT72AY1.

We hypothesize that in the bi-substrate enzyme *Nb*UGT72AY1, asymmetric cooperativity of substrates causes SI, and thus at high acceptor concentration, formation of the non-productive conformer predominates (Fig. 7). This mechanism extends the models of bi-substrate enzyme reactions proposed by Cleland[15]. A mechanism, termed iso-random bi bi, is similar to the asymmetric cooperativity mechanism and has been proposed for the catalysis of a protein kinase[40]. Although the mechanism was originally postulated to be formally random sequential binding, at limited amounts of acceptor substrate there appears to be a preferential binding order, with donor ATP binding first. At higher acceptor concentrations, binding of the acceptor is favoured and when the donor ATP subsequently binds, an unproductive enzyme-substrate conformer is formed that can reversibly convert to a productive ternary complex.

Cooperativity is a manifestation of a communication network in proteins[41]. In the case of bi-substrate enzymes, cooperativity means an interplay between the two binding events in the active site, one from each substrate (heterotrophic cooperativity)[16]. However, monomeric bi-substrate enzymes are also subject to homotropic allosteric regulation[42]. The mnemonic model and the ligand-induced slow transition model (LIST) are frequently used to explain non-hyperbolic kinetics in monomeric enzymes[41]. In agreement with the mnemonic and LIST models, various protein complexes of *Nb*UGT72AY1 were resolved by X-ray crystallography. The complexes demonstrate that the binding of the substrates is sequential random, which is a premise for the third hypothesis of non-Michaelis-Menten kinetics, kinetic cooperativity in monomers[43].

*Nb*UGT72AY1 exhibits homotropic cooperativity for both the acceptor and donor substrates, whereas the Hill coefficient for UDPG is dependent on the scopoletin concentration used, indicating also heterotropic cooperativity (Fig. 5). Since the binding of the ligands occurs sequential random via two different pathways, the reaction rate of the overall reaction is best described by an equation with two Hill coefficients (equ. 2; Supplementary Note 13). In the case of UDPG, the Hill coefficients ($h$) for both complex formation pathways are identical, i.e. the cooperativity is symmetrical and independent of the order of ligand binding (equ. 3; Supplementary Note 13). This results in a sigmoidal curve or even a hyperbola if h = 1. In the case of scopoletin, however, the Hill coefficients leading to the alternative complexes differ ($n \neq x$), i.e. the cooperativity in the pathways is different and therefore asymmetrical (equ. 2; Supplementary Note 13). This leads to a sigmoidal curve with SI kinetics. By asymmetric cooperativity we mean the unequal homotropic influence on ligand binding in the two branches of a sequential random binding system ($n \neq x$). One consequence of the inequality of the Hill coefficients is SI.

Since all biocatalysts except isomerases and around 60% of the known industrially important enzymes[16], require at least two substrates, the postulated mechanism could also apply to many more enzymes. However, as long as the cooperativity in the two pathways is identical, a Michaelis-Menten relationship results ($n = x$ in equ. 2; Supplementary Note 13) which obscures the mechanism. *Nb*UGT72AY1 is another rare example demonstrating that cooperativity does not require multiple homotropic ligand binding events or multimeric assemblies. Among the models that attempt to explain cooperativity in monomeric enzymes, the alternative pathway for a two-substrate enzyme comes closest to our results[41]. According to this model, random addition of substrates can result in cooperative behaviour if a particular pathway of substrate addition is kinetically preferred (kinetic cooperativity)[43,44]. Our results go beyond this as they provide a hypothesis for the molecular mechanism of kinetic cooperativity and show that effectors influence random ligand binding by competitive inhibition, thereby suppressing the formation of the non-productive ternary enzyme complex. The new model consolidates two classical models (random and sequential ordered) for substrate binding in a bi-substrate enzyme.

Although the concepts of cooperativity and allostery have different origins and meanings, they are linked by a common molecular principle[45]. Cooperativity occurs in a molecular binding system if a

receptor molecule is able to bind more than one ligand[16]. The binding of the ligands to different sites on the receptor does not constitute mutually independent events. Binding is cooperative when the attachment of the first ligand to the receptor alters the binding properties for the second ligand. Allostery is defined as the regulation of a protein by binding an effector molecule at a site other than its functional site[46]. Nature exploits allosteric regulation for signal transduction, activation of enzymes and thus regulation of metabolism, motor work and control of transcription. Both concepts are united by receptor-ligand binding, which is accompanied by a conformational change in the receptor that alters the function of the receptor-ligand complex. Allosteric enzymes are therefore cooperative and do not obey Michaelis-Menten kinetics, but instead exhibit sigmoidal kinetics that can be described by the Hill equation. The allosteric descriptor $\alpha$ (allosteric intrinsic efficacy of a ligand) was introduced to combine three viewpoints of allostery namely thermodynamics, free energy landscape of population shift, and structure[46]. Since the considerations carried out for receptors are also valid for enzymes, the assumption that a measured enzymatic activity is proportional to the proportion of enzyme (E) in the activated state (E') leads to the equation $v/v_{max} = L/(1 + L)$, where $L = [E']/[E]$ (Supplementary Note 14)[46]. The allosteric response can then be formulated as $v/v_{max} = \alpha L/(1 + \alpha L)$ (Supplementary Note 14). From comparison with the Hill and Michaelis-Menten equations, we can derive $\alpha L = ([S]/K_S)^n$ and $\alpha = ([S]/K_S)^{n-1}$. Thus, there is a relationship between the Hill coefficient and the allosteric intrinsic efficacy factor, with no allosteric effect observed when n = 1, as $\alpha = 1$. The Hill coefficient is a measure of cooperativity in a binding process but it is not an accurate measure of the number of binding sites[47]. Due to its relationship with $\alpha$, the Hill coefficient is rather a parameter describing the extent of protein population shift as a function of ligand concentration and its dissociation constant.

Numerous human enzymes e.g. important drug-metabolizing enzymes such as monooxygenases, glycosyltransferases and kinases show SI, a widespread phenomenon, but one that is still poorly understood[2]. SI of enzymes is one of the major obstacles in industrial biocatalysis, although in many cases SI can be mitigated by increasing the $K_m$ values of the inhibiting substrate by protein engineering without affecting $v_{max}$[48]. Our results demonstrate that competitive inhibitors can achieve this effect by preventing the binding of the substrate in the conformer, which leads to the inactive complex. Furthermore, since an imbalance in the binding properties of the substrates of a bi-substrate enzyme leads to SI, the results show that weakening the affinity for both the acceptor (T145L) and donor (W350A) substrates attenuates SI, findings that may result in the engineering of improved biocatalysts in the future.

The physiological effects of SI of NbUGT72AY1 and the opposing effect of (apo)carotenoids are difficult to assess. In vitro tobacco plants treated with scopoletin via the roots produced more scopolin[24] in the presence of apocarotenoids and in this study, we showed that after chemical induction of scopoletin formation more scopolin was generated in transgenic tobacco plants that overproduced β-carotene. These findings point to a previously unreported mode of action of β-carotene. Furthermore, they reveal a previously unknown mechanism of enzyme regulation through reactivation of a substrate-inhibited enzyme by metabolites of the plant's specialized metabolism. This provides new impetus for further research into the physiological functions of (apo)carotenoids and other plant secondary metabolites and can contribute to improving plant resilience and food quality in the near future.

## Methods

### Cloning of NbUGT72AY1 and StUGT72AY2 mutants

Cloning of NbUGT72AY1 from Nicotiana benthamiana (accession number; gene MT945401; protein UHH90560) and generation of NbUGT72AY1-I86V_F87I, -H390Y, -N27D, -I86V, -F87I and NbUGT72AY1-Chimera V154-F209/I152-C204 (Tobacco-Potato-Tobacco) and StUGT72AY2-V83I_I84F, -N27D, and -Y389H were performed according to[8–10]. Genewiz, Leipzig, Germany (www.genewiz.com) synthesized StUGT72AY2 (PGSC0003DMG401004500; accession number gene XM_015308592; protein XP_015164078) and NbUGT72AY1-Chimera A49-E63/S49-K60. The genes were ligated into the pGEX-4T-1 vector. NbUGT72AY1-V184M, -K171E, -S191N, -L187M_L188M, -R179Q, -T145L, -K157Q, -E165D, -E168Q, -Y317F, -Y163H, -F155I, -E160Q, -W350A, and -N27A, and StUGT72AY2-Chimera S49-K60/A49-E63 were generated by site-directed and chimera mutagenesis according to[9].

### Protein production for kinetics studies

Protein expression was executed using E. coli BL21(DE3) pLysS cells. Following an overnight pre-culture at 37 °C in Luria-Bertani medium supplemented 100 μg/mL ampicillin and 34 μg/mL chloramphenicol, the pre-culture was mixed with the main culture in a 1:100 ratio, both containing the respective antibiotics. This mixture was incubated at 37 °C until the OD600 reached 1 in a baffled flask. The cells were collected via centrifugation and stored at -80 °C. Recombinant fusion proteins featuring an N-terminal GST tag were purified by employing Novagen (Darmstadt, Germany) GST Bind Resin, adhering to the manufacturer's instructions. After resuspension, sonication was used to disrupt the cells. Post-centrifugation, the crude protein extract was left to incubate overnight at 4 °C with the resin to facilitate binding of the GST fusion protein, and it was subsequently eluted using GST elution buffer containing reduced glutathione. The quality of the purified proteins was verified through SDS–PAGE (Supplementary Fig. 40), and the protein concentration was determined using Roti-Nanoquant (Carl Roth, Karlsruhe, Germany) in 96-well microtiter plates as per the manufacturer's instructions. Absorbance was measured at 450 nm and 590 nm using a CLARIOstar plate reader (BMG Labtech, Germany).

### Determination of UDP-α-glucose glucohydrolase and glucosyltransferase activity of NbUGT72AY1 and effects on activity by effectors

The assay consisted of 0.5 μg purified NbUGT72AY1, 0-1000 μM retinol, β-apo-8'-carotenal, β-carotene, lycopene, zeaxanthin, or lutein (effectors) in 50 mM Tris–HCl (pH 7.5) in a final volume of 100 μL. After application of 0–1000 μM scopoletin (acceptor substrate), the reaction was started by adding 100 μM UDP-glucose (donor substrate) and incubated at 40 °C for 10 min. Inactivation solution (12.5 μL of 0.6 M HCl) was directly added to stop the reaction, followed by 12.5 μL neutralization solution (1 M TRIZMA base) to regulate pH. UDP-Glo™ Glycosyltransferase Assay (Promega, Walldorf, Germany) was used to measure free UDP concentrations according to the manufacturer's instruction. The UGT reactions (5 μL) were mixed with 5 μL detection reagent (UDR). Luminescence was measured with CLARIOstar plate reader after 60 min incubation in the dark and at room temperature. Triplicates were performed for each measurement.

### UDP-Glo assay based analysis of the bi-substrate enzyme NbUGT72AY1 after pre-incubation with β-carotene or scopoletin

The assay consisted of 0.5 μg purified NbUGT72AY1, 0–100 μM scopoletin (acceptor substrate) in 50 mM Tris–HCl (pH 7.5) in a final volume of 100 μL. The reaction was started by adding 0–100 μM UDP-glucose (donor substrate) and incubated at 40 °C for 10 min. To investigate the time-dependent alteration in enzyme activity by the acceptor substrate and effector, 0–100 μM scopoletin or 0–1000 μM β-carotene were pre-incubated with the enzyme for the indicated time intervals of 0–900 sec before the start of the reaction. Enzyme activities were calculated by measuring of free UDP using the UDP-Glo™ assay. Selected reactions were also analysed by LC-MS according to[9].

## Protein modeling

Ligand docking (β-carotene, β-apo-8′-carotenal, and retinol) was performed by AutoDock Vina[49] (http://vina.scripps.edu/) using Chimera[50] (v.1.17.1, https://www.cgl.ucsf.edu/chimera/) for visualization. Default values were used.

## Protein expression, production and purification for crystallography

NbUGT72AY1 was cloned into the pGEX6P-1 expression vector (GE Healthcare, Solingen, Germany). The plasmid was then transformed into BL21 (DE3) cells for expression. Cells were grown in LB broth media containing ampicillin (100 µg/mL) at 37 °C until an OD600 of 0.6 was reached. Protein expression was induced by adding 200 µM isopropyl β-D-1-thiogalactopyranoside (IPTG), and the cultures were incubated at 20 °C overnight. Cells were harvested by centrifugation at 6500 $g$ for 10 min. The cell pellet obtained from one liter culture was re-suspended in 25 mL lysis buffer (50 mM Tris-HCl (pH 7.5), 250 mM NaCl, 3 mM DTT, and 0.5% (v/v) TritonX-100) and lysed by sonication. Cell debris was removed by centrifugation at 50,000 $g$ for 30 min, and protein was purified from the obtained supernatant using Glutathione Sepharose 4B resins (GE Healthcare) as shown before in ref. 51. The N-terminal GST tag of NbUGT72AY1 was removed by overnight incubation with PreScission Protease (GE healthcare) at 4 °C. After the GST tag cleavage, the protein was further purified on a HiLoad16/60 Superdex 200 prep-grade gel filtration column (GE Healthcare) using a buffer containing 20 mM HEPES (pH 7.5), 150 mM NaCl, and 3 mM DTT. The purified protein was concentrated to 8 mg/mL and stored at -80 °C.

## Crystallography

NbUGT72AY1 was crystallized using sitting drop vapor diffusion method. Apo-NbUGT72AY1 and NbUGT72AY1 were co-crystallized with various ligands such as 2 mM UDP (Sigma), 1 mM UDP-2-deoxy-2-fluoro-D-glucose (BOC Sciences), 1 mM scopoletin (Sigma), 0.5 mM β-carotene (Sigma), and 1 mM retinol (Sigma) by equilibrating 1.0 µL of protein (8 mg/mL) mixed with 1.0 µL of reservoir solution (Supplementary Data 1). All crystals grew within 7 days at 23 °C. Obtained crystals were soaked with the ligands 0.5 mM β-carotene or retinol for 48 hours. Crystals were then cryoprotected using a solution resulting from mixing equal volumes of 50% glycerol with 50% well solution containing 0.5 mM ligand, and flash-cooled in liquid nitrogen. All data were collected at 100 K at the beamlines Proxima 1 and Proxima 2 A at the SOLEIL Synchrotron (France), using a EIGER-X 16 M and EIGER-X 9 M detectors, respectively (proposal numbers 2021095, 20210932, 20220373, and 20221065). The data were processed in XDS[52]. Additional data processing was done for 9LRJ using STARANISO webserver[53,54] (https://staraniso.globalphasing.org) to improve the data statistics in the high resolution shell. The crystal structures were determined by molecular replacement using BALBES[55] with the TcCGT1 structure (PDB 6JTD) as search model. The refined apo-NbUGT72AY1 model was then used as template for molecular replacement to determine the structures of NbUGT72AY1 with various other ligands. All the structures were initially refined using Phenix Refine[56]. The low resolution structure (9LRJ; 3.10 Å) was additionally refined using the Lorestr pipeline in ccp4[57], which inspects the data for twinning, checks for the best scaling option to implement for refinement, and uses H-bond external restraints information obtained from homologous structures (PDB: 8J2V and 8J2Z) using proSMART[58]. The refinement incorporates homology-derived restraints to improve the geometry of the model, followed by jelly body refinement. The pipeline evaluates both standard and least-squares scaling methods, opting for the one that yields a lower Rfree value. It also determines the parameters for modeling the solvent, such as VDWProb 1.10, IONProb 0.80, and RSHRink 0.80. All other structures were processed by the PDBredo[59] pipeline which uses the above mentioned parameters for

low resolution structures using Refmac5[58,60]. The structures were manually inspected using Coot[61] (Supplementary Data 1). The figures were drawn using PyMOL (www.pymol.org/pymol.html).

## Differential scanning fluorimetry (DSF)

NbUGT72AY1 (10 µM) was incubated with a combination of various ligands such as 100 µM scopoletin, UDP-2-deoxy-2-fluoro-D-glucose, and retinol. SYPRO Orange dye (Thermo Fisher, Waltham, MA) was added at the final concentration of 2x to the samples. The reaction was setup at 25 µL in a size exclusion buffer, and the assay was performed on CFX96 Real-Time System (BIO-RAD, Hercules, CA, USA) thermal cycler, as reported previously[62]. The data were plotted and final figure drawn using Graphpad 9.0 (Prism; www.graphpad.com).

## Size-exclusion chromatography coupled to multi-angle light scattering (SEC-MALS)

The SEC-MALS data for NbUGT72AY1 were obtained after passing through the Superdex 200 30/300 column (GE Healthcare) in a buffer containing 20 mM HEPES, 200 mM NaCl, and 3 mM DTT pH 7.5. The protein output was then passed through Agilent HPLC following DAWN MALS detector (Wyatt Technology). Data were analysed using ASTRA software provided by the company.

## Molecular dynamics (MD) simulations

The initial coordinates of the MD simulations were obtained from complex V (PDB 9J9K). Hydrogens missing from both the protein and ligand atoms were added, and the hydrogen bond network of side chains was optimized using the Protein Preparation Wizard tool under physiological conditions (pH 7.4) to ensure the correct protonation states (Schrödinger Release 2022-1, Maestro, Schrödinger, LLC, New York, NY, 2022)[63,64]. The prepared protein was solvated in a cubic box with a padding of 15 Å, using the TIP3P water model[65], and neutralized with a proper number of sodium and chloride ions to reach a salt concentration of 0.154 M. The system was built with the Leap software implemented in AMBER (The Amber Molecular Dynamics Package, at http://ambermd.org)[66] and AmberTools22. The protein was parametrized using the ff14SB force field[67]. Ligands (scopoletin and UDPG) topology, parameters and partial charges were retrieved using antechamber and parmcheck2 tools, assigning parameters from GAFF force field[68]. ACEMD[69] (Acellera, version 3.7) was used for the MD simulations with periodic boundary conditions. The systems were initially equilibrated through a 1000 conjugate gradient step minimization to reduce clashes induced by the system preparation between protein and water atoms and then equilibrated with a 4 ns MD simulation in the isothermal–isobaric conditions (NPT ensemble), employing an integration step of 2 fs. Initial restraints of 5 kcal mol$^{-1}$ Å$^{-2}$ were gradually reduced in a multistage procedure over the 4 ns: 2 ns for all protein atoms other than Cα atoms, 4 ns for the protein Cα atoms, and 4 ns for all the ligand atoms. The temperature was maintained at 298 K using a Langevin thermostat[70] with a low damping constant of 1 ps$^{-1}$, and the pressure was maintained at 1.01325 atm using a Montecarlo barostat. The M-SHAKE algorithm[71] was used to constrain the bond lengths involving hydrogen atoms. Long-range Columbic interactions were handled using the particle mesh Ewald summation method[72] with grid size rounded to the approximate integer value of cell wall dimensions. The cutoff distance for long-term interactions was set at 9.0 Å, with a switching function of 7.5 Å. Then, we ran three independent replicates for each equilibrated system of 1 µs unrestrained MD simulations in the canonical ensemble (NVT) with an integration time step of 4 fs. The temperature was set at 298 K, by setting the damping constant at 0.1 ps$^{-1}$. The trajectory was saved each 0.1 ns. Root Mean Square Deviation (RMSD) and Root Mean Square Fluctuation (RMSF) values were computed for all the simulated trajectories using an in-house Python (v3.11) script based on MDAnalysis (v2.2.0)[73]. The initial coordinates of complex V (PDB: 9J9K) were used

as the reference for the structural alignment. Chi1 (χ1, between N-CA-CB-CG) and chi2 (χ2, between CA-CB-CG-CD1) torsion angle analysis of Trp350 were performed with MDtraj (v1.9.7)[74]. The three replicates of each system were merged into a single trajectory for the dihedral analysis. Visualization of all data was carried out with the Matplotlib Python library[75] (Supplementary Table).

## Hydrogen/deuterium exchange mass spectrometry (HDX-MS)

HDX-MS experiments probing ligand binding to *Nb*UGT72AY1 were conducted as described previously[9,24]. Ligands (scopoletin, β-carotene, apo-8'-carotenal, retinol) were employed as 50 mM concentrated stock solutions dissolved in DMSO. Prior HDX-MS, 196 µL of purified *Nb*UGT72AY1 was mixed with 2 µL of ligand stock solution or solvent to reach final concentrations of 40 µM (*Nb*UGT72AY1) and 500 µM (ligands), respectively, and incubated for 5 min at ambient temperature. Further preparation of exchange reactions for HDX-MS was aided by a two-arm robotic autosampler (LEAP Technologies). A 7.5 µL protein solution was mixed with 67.5 µL of D$_2$O-containing buffer (20 mM Tris-Cl pH 7.5, 2% (v/v) DMSO), which also contained 500 µM of ligands to prevent their dilution during HDX. After incubation for 10/30/95/1,000/10,000 sec at 25 °C, the HDX reactions were added to 55 µL of pre-dispensed quench solution (400 mM KH$_2$PO$_4$/H$_3$PO$_4$, 2 M guanidine-HCl, pH 2.2) kept at 1 °C and 95 µL of the resulting mixture injected into an ACQUITY UPLC M-class system with HDX technology (Waters). *Nb*UGT72AY1 was digested online with porcine pepsin and separated by reversed-phase HPLC followed by mass spectrometric analysis. Peptides were identified from the non-deuterated samples with the ProteinLynx Global Server 3.0.1 (Waters) software. For quantification of deuterium incorporation with DynamX 3.0 (Waters), peptides had to fulfil the following criteria: minimum intensity of 5000 counts; maximum length of 30 amino acids; minimum number of products of three; maximum mass error of 25 ppm; retention time tolerance of 0.5 min. After automated data processing by DynamX 3.0, the mass spectra were manually inspected and, if necessary, peptides omitted, e.g., in case of low signal-to-noise ratio or presence of overlapping peptides. HDX-MS raw data are supplied in Supplementary Data 6[76].

## Isothermal titration calorimetry (ITC) measurements

ITC measurements were conducted on a MicroCal PEAQ-ITC (Malvern Panalytical, Malvern, UK). Assay buffer consisted of 20 mM sodium phosphate buffer, 150 mM NaCl, 3 mM DTT, pH 7.3. Purified *Nb*UGT72AY1 was dialysed against the assay buffer and spin-concentrated. Scopoletin ligand stocks were prepared in DMSO to a concentration of 50 mM and the sample was kept under continuous flushing of nitrogen gas to remove the DMSO and then dissolved in 1% Triton X-100. The assay buffer was degassed before use in control titrations and preparation of the ligand and target sample. The target sample (in the cell) consisted of 25 µM *Nb*UGT72AY1 and the scopoletin ligand samples (in the syringe) were prepared to a concentration of 250 µM with the presence of 0.005% Triton X-100 in both the samples. Titrations of ligand into assay buffer were used as control. Measurements were set up as follows: 18 µM *Nb*UGT72AY1 (in cell), 400 µM ligand (in syringe), with 19 injections (0.4 µL + 18 × 2.0 µL), with reference power 10 µcals/s, stirring speed 750 rpm, temperature 25 °C. Data were analysed using the MicroCal PEAQ-ITC analysis software.

## Circular Dichroism (CD) measurements

CD measurements were conducted on a J-1000 series CD spectrophotometer (Jasco Inc., USA). Purified *Nb*UGT72AY1 in 20 mM sodium phosphate buffer, pH 7.5 and 100 mM NaF was used as buffer for preparation of samples. Scopoletin and β-carotene were prepared as ligand stocks at 50 and 4 mM in DMSO respectively. Final samples for measurement consisted of 10 µM *Nb*UGT72AY1 in various combinations with and without 100 µM ligand. Measurements were carried out at 20 °C in a 1 cm cuvette. As control, buffer and buffer-ligand measurements were subtracted. The results are an average of 3 repeats each with 10 scans taken with the wavelength ranging from 190 to 250 nm, buffer and baseline were corrected and the data presented as molar residue ellipticity (mdeg).

## In planta experiment

Tobacco (*Nicotiana tabacum* cultivar Xanthi) wild type and transgenic *Daucus carota* lycopene β-cyclase 1 (*Dc*LCYB1)-expressing line 14[37,38] plants were grown under 16-hr photoperiod as described by ref. 10. *Dc*LCYB1-expressing line 14 was kindly provided by Claudia Stange, University of Chile, who generated the line[38]. Leaves of six-week old plants were briefly immersed in a 2,4-dichlorophenoxyacetic acid/polyethylene glycol 1500 (PEG)-solution and PEG solution, respectively, according to[77]. After 24 h, leaves were harvested and frozen in liquid nitrogen. Whole leaves were weighed, ground and extracted four times with 10 mL methanol. As internal standard 40 µg 4-methylumbelliferyl β-D-glucopyranoside was added. Extracts were concentrated to dryness with a rotary evaporator, resuspended in 2 mL 80% (v/v) methanol, concentrated to dryness with a SpeedVac and finally dissolved in 75 µL 80% (v/v) methanol for LC-MS analysis. LC-MS analysis was conducted according to Sun et al., 2023[24].

## Reporting summary

Further information on research design is available in the Nature Portfolio Reporting Summary linked to this article.

## Data availability

Accession code for *Nb*UGT72AY1 at GenBank UHH90560.1 (protein), MT945401.1 (mRNA) and *St*UGT72AY2 at Genbank XP_015164078.1 (protein), XM_015308592.1 (mRNA). All X-ray crystal structures generated in this study are deposited at pdb (https://www.rcsb.org/structure/), provisional IDs: 8J2Z, 8J31, 8J2V, 8J2U, 9J9K, 9LRJ). MD trajectories and related files (topology, parameter, and coordinates) are available at Zenodo (https://zenodo.org/records/14001338). The mass spectrometry proteomics data have been deposited to the ProteomeXchange Consortium via the PRIDE[78] partner repository with the dataset identifier PXD056344. Source data are provided with this paper as a Source Data file. Source data are provided with this paper.

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

## Acknowledgements

We are deeply thankful to Juan C. Moreno and Salim Al-Babili for providing the transgenic DcLCYB1-expressing tobacco plants. The lines were generated in the laboratory of Claudia Stange at the University of Chile, Santiago, Chile. We sincerely thank her for providing us with the mutant lines. We are grateful to Johannes Buchner for the use of the CD instrument. We acknowledge SOLEIL for provision of synchrotron radiation facilities and would like to thank P. Legrand, S. Sirigu, M. Savko and B. Shepard for assistance in using the beamlines PROXIMA 1 and PROXIMA 2 A. For computer time, this research also used the resources of the KAUST Supercomputing Laboratory, and the Bioscience Core Lab and the Imaging and Characterization Core Lab at King Abdullah University of Science & Technology (KAUST) in Thuwal, Saudi Arabia supported experimental research. The German Research Foundation grant SCHW 634/34-1 provided financial support for the research of WGS. The King Abdullah University of Science and Technology (KAUST) supported the research by U.S.F.H. and S.T.A.

## Author contributions

Conceptualization: W.G.S., T.H.; Methodology: J.L., U.F.S.H., T.D.H., E.K., W.S., A.N., A.D.P., C.K.; Investigation: J.L., U.F.S.H., T.D.H., E.K., G.S., W.S., A.N., D.A.M.C.; Visualization: J.L., U.F.S.H., E.K., W.S., A.N., A.D.P., T.D.H., D.A.M.C.; Funding acquisition: W.G.S., S.T.A., W.S., C.S.; Project administration: W.G.S., T.H.; Supervision: T.H., W.G.S., S.T.A.; Writing – original draft: W.G.S., J.L., E.K., G.S., W.S., U.F.S.H., A.D.P.; Writing – review & editing: W.G.S., S.T.A., C.S., C.K., F.A.G.

## Funding

## Competing interests

The authors declare no competing interests.
