## [Transparent Peer Review file · Nature Communications]

β -Carotene alleviates substrate inhibition caused by asymmetric cooperativity

Corresponding Author: Professor Wilfried Schwab

Version 0:

Reviewer comments:

Reviewer #1

(Remarks to the Author)

Liao, J et al., reports the molecular mechanism of substrate inhibition of tobacco glucosyltransferase NbUGT72AY1 using crystal structures, enzyme kinetics and other biophysical approaches. Based on the structures of enzyme-substrate complexes, authors conclude that this enzyme does not follow the classical enzyme-catalyzed sequential bi-substrate reaction but suggests asymmetric cooperativity as an alternative pathway. As the structure coordinates is available in PDB, a closer look into the ligand density and model quality, especially for the critical residues the authors are describing the overall mechanism, is a major concern to me. Please see my comments below as it will help the authors to revisit the model building and structure refinement for structure based conclusion. In my opinion, the manuscript is not suitable for publication in Nature communication in its current stage.

It is advisable to check the electron density of the ligands and B-factors (U2F and T83) in the deposited structure (PDB ID 8J2S). The B-factors of all atoms in both the ligands are 46.5 \AA^2 . Authors should check all the structures described in this manuscript thoroughly and perform proper refinement. Interestingly, the density for the suggested helical transition upon ligand binding (46-59) is very poor and misleading. The details of the crystallographic refinement need to be described in the manuscript. Authors should use refinement strategies used for low resolution structures and define the refinement protocol in the manuscript.

PDB ID 8J31: Protein crystallized in presence of beta carotene, but the ligand is not present in the structure. The side chain flipping cannot be considered a major point of discussion without having the ligand density and its interaction with the respective side chain and local structural rearrangement. Please report in the method section whether the ligands were used in the crystal freezing steps.

PDB ID 8J2V: Residues 310-326 is disordered/out of density/no density, but the authors are making important conclusions based the structural rearrangement upon ligand binding (scopoletin), the density of which is also misleading. Also, the suggested helical transition based on the maps is not suggestive as for 8J2S. Please generate omit maps for ligands. Also, the placement of sulfate ion (A/502) is not supported by the density.

PDB ID 8J2T: It is critical to provide omit maps for the ligand density. There is an unaccounted positive density in the active site next to Q353, which seems to be the ligand density. Also, the rotamer of for critical residue W350 seems to be incorrect. The authors should revisit the maps to better understand the mechanism.

Authors should provide omit maps for the ligands from the crystallographic studies. Specifically, the resolution for 8J31 is poor with very high R-factors. B-carotene interaction distances from the protein side chains (T141, H139 and Y145) seems to be very high. Authors should provide rational and include the electron density maps to support Sup. Fig. 13.

H18, D118, W350 side chain density should be shown as omit maps for Sup. Fig 10 as the interaction distance is large and seems to be changing based on ligands. Also, Sup. Fig. 10/VI-F: please check H18 and D118 side-chain conformation is supported by the electron density. Also, authors should provide omit maps for UDP2FG-W350 stacking. Authors should include the crystallization condition. Also, please mention whether the corresponding ligands have been used in the cryoprotective solution or not. The presence or absence of ligands in the crystal freezing stage is critical in this case as the affinity is low and the structural analysis/conformational change would be wrongly interpreted.

The use of software should be properly cited: Phenix, COOT, CCP4, chimera, PyMOL, XDS,

Low molecular weight contamination is evident in SDS-PAGE gel provided by the authors. The sample purity is critical for ITC and CD and other biophysical experiments, I recommend to further purify protein sample and redo ITC and CD experiments. The ITC isotherm for buffer control is not ideal and should be repeated along with experimental titration to obtain reasonable ITC isotherm.

Reviewer #2

(Remarks to the Author)

In the submitted manuscript, Liao et al. study the mechanism of substrate inhibition on the enzyme NbUGT72AY1, a glycosyltransferase from *Nicotiana benthamiana*. The authors took advantage of their finding that retinol and β -carotene, while acting as a competitive inhibitor of the acceptor substrate, scopoletin, appeared to strongly attenuate substrate inhibition resulting from excess acceptor substrate. The authors were able to co-crystallize the enzyme with or without substrates or products in the presence or absence of effectors (including retinol) and obtain X-ray crystal structures that they associated with productive or non-productive complexes of the enzyme. From these structure, the authors propose a mechanistic model to explain substrate inhibition in NbUGT72AY1, in which productive complex formation occurs when the donor substrate (UDP-glucose) binds first followed by the acceptor substrate (scopoletin), whereas a non-productive complex forms when the acceptor substrate binds first followed by the donor substrate. The authors propose that effectors like retinol and β -carotene can block the acceptor-binding site with minimal conformational change of the protein until the donor can bind and then the effector can be competitively replaced by the acceptor substrate.

As it is estimated that substrate inhibition occurs in roughly 20% of enzymes, insights into substrate inhibition are generally important in enzymology, and of particular interest to industrial biocatalysis and pharmacology with respect to enzyme targets.

The authors state that to date, no structural evidence supports current models of unproductive enzyme-substrate complexes being involved in substrate inhibition, and thus the work reported in this manuscript is important in providing such evidence. (The authors citation of references 1-3 for this lack of evidence refer to some papers that are over 10 years old, however, and the most recent citation is a commentary.)

It's not clear to this reviewer, considering that retinol and β -carotene are competitive with scopoletin (and should occupy to the same acceptor-binding site), why incubation with excess effectors does not also lead to a non-productive complex when they bind before the donor. Is it because these effectors do not cause as large a conformational change as scopoletin does when it binds first, or is it because these effectors do not bind as tightly? The lack of X-ray crystal structures including either retinol or β -carotene as a ligand leaves this ambiguous.

Page 11, line 20: I do not agree that the lack of observation of scopoletin in more than one binding site in the X-ray crystal structures refutes the hypothesis that it can bind in more than one site.

Overall, the work is well done. Methodology is sound and enough detail is provided.

Reviewer #3

(Remarks to the Author)

This article proposes a molecular mechanism of substrate inhibition of a tobacco glycotransferase. The authors utilised x-ray crystallography to offer snapshots of the molecular reaction mechanism involving a bi-substrate enzyme. The also employ HDX-MS and molecular dynamics to shed light on the conformational changes occurring substrate binding in active and inhibited protein complexes. Overall, this appears to be a comprehensive hat aims at explaining substrate inhibition in multi-substrate enzyme. However, I feel that the study can be improved following some suggestions below:

1. Whilst HDX-MS constitutes a small part of the article, the presentation of data lacks some information. There is no information about statistics of the experiments. For instance how many technical and biological replicates have been performed? In supp figures 7 and 22, it is hard to judge whether or not the differences in deuterium uptake are statistically significant or not. I would suggest the authors to use the wood's plots. There are specialised softwares (e.g. Deuterios) that can carry out such analyses.
2. What is the back exchange in the HDX-MS experiments? From Supp Figures 7 and 22 it appears that HDX differences are small (e.g. from -20 to 20). That may indicate high back exchange (>30%). It will be good if the authors can comment on this.
3. I would recommend the authors to include information about the peptide coverage, redundancy they obtained for each experimental condition (see recommendation from the community white paper that the authors cite here, Masson et al). Suppl. Table 6 provides such information, but this needs to be expanded.
4. Moreover, the redundancy appears a bit low, especially for a soluble protein. It would be good if the authors provide information regarding the optimisation steps that they took to achieve such sequence coverage.

Version 1:

Reviewer comments:

Reviewer #1

(Remarks to the Author)

Although, the authors have made substantial improvement in the revised manuscript, I still have concerns, which needs to be addressed for the manuscript to be considered for publication in Nature Communications.

I'm concerned about the placement of scopoletin in 8J2V. Although, the omit map has been provided, the fitting is not clear as the FoFc density map does not seem to be supporting the full ligand geometry. Authors should check the fitting and perform occupancy refinement and provide clear picture for the occupancy of the ligand. Also, it is highly unlikely that the ligand has only two interacting atoms more than 4Å apart. Please also provide the LigPlot for scopoletin. I have attached FoFc map for scopoletin at 0.7σ and 0.6σ.

The lack of B-carotene (8J31) and model error (residue 53-63) and poor resolution suggest over interpretation. The authors should try rMMS technique to obtain protein crystals in new condition in presence of very high ligand concentration. Without the ligand density, this structure is not telling anything significant.

There are several Ramachandran outliers and clash score is very high in 9J9C. CC1/2 and I/sigI is very low as compared to other structures. Authors should improve the structure quality. Authors should take advantage of alphaFold model.

Authors should pay attention in Table 1. The units are missing at several places. Also, the Ligand B-factor should be included.

Please mention the reference model and parameters used in Lorestr refinement.

PYMOL should be replaced with PyMOL.

Reviewer #2

(Remarks to the Author)

All of my comments have been addressed in the revision.

Reviewer #3

(Remarks to the Author)

The authors have carried out a poor review of the article barely addressing any comments from HDX-MS experiments. They could have included the wood's plots for all timepoints used. Also, without assurances about the back-exchanges which appears to be large and lack of optimisation it is still questionable if the data are fully usable. Unless these are done, I cannot recommend publication of the article.

Version 2:

Reviewer comments:

Reviewer #1

(Remarks to the Author)

Authors have satisfactorily addressed all of my comments in the revised version of the manuscript and it should be accepted for publication in Nature communications.

Reviewer #3

(Remarks to the Author)

The authors have added the recommended wood's plots in the revised version. I am happy to recommend publication.

Thank you very much for the comments of two reviewers. The comments are very helpful and contribute to improving the manuscript.

REVIEWER COMMENTS

Reviewer #1 (Remarks to the Author):

Liao, J et al., reports the molecular mechanism of substrate inhibition of tobacco glucosyltransferase NbUGT72AY1 using crystal structures, enzyme kinetics and other biophysical approaches. Based on the structures of enzyme-substrate complexes, authors conclude that this enzyme does not follow the classical enzyme-catalyzed sequential bi-substrate reaction but suggests asymmetric cooperatively as an alternative pathway. As the structure coordinates is available in PDB, a closer look into the ligand density and model quality, especially for the critical residues the authors are describing the overall mechanism, is a major concern to me. Please see my comments below as it will help the authors to revisit the model building and structure refinement for structure based conclusion. In my opinion, the manuscript is not suitable for publication in Nature communication in its current stage.

It is advisable to check the electron density of the ligands and B-factors (U2F and T83) in the deposited structure (PDB ID 8J2S). The B-factors of all atoms in both the ligands are 46.5 \AA^2 . Authors should check all the structures described in this manuscript thoroughly and perform proper refinement.

Response: Thanks for spotting this glitch in refinement. We now used the lorestr program in ccp4 for additionally refining the 3.1 \AA resolution structures 8J2S (now redeposited as 9J9C).

Interestingly, the density for the suggested helical transition upon ligand binding (46-59) is very poor and misleading.

Response: The corresponding region in apo and beta carotene soaked structures appears to be highly disordered compared to structures to which scopoletin is bound. We have included the omit map for the helix transition region (residues 45-60) in the scopoletin bound structure (Supplementary Data 7C). Additionally, we have included a figure showing the structure colored by B-factor for the scopoletin bound structure (8J2V) and the apo/beta carotene soaked structures (8J2Z, 8J31, respectively; Supplementary Data 7B). We also added the omit density for scopoletin in the scopoletin-bound structure 8J2V (Supplementary Data 7D).

The details of the crystallographic refinement need to be described in the manuscript. Authors should use refinement strategies used for low resolution structures and define the refinement protocol in the manuscript.

Response: In addition to the initial refinement with Phenix, we subjected 5 of the 6 structures to a refinement of PDB redo (which also has predefined parameters for structures below 3 \AA). The final structure, namely 8J2S->9J9C, was directly processed by the ccp4 program lorestr (due to the presence of the high total number of ligands). This is now clarified in the revised version.

PDB ID 8J31: Protein crystallized in presence of beta carotene, but the ligand is not present in the structure. The side chain flipping cannot be considered a major point of discussion without having the ligand density and its interaction with the respective side chain and local structural rearrangement.

Response: We have made several attempts to co-crystallize and soak with beta carotene, but our electron density maps failed to show electron density for this molecule. Thus, we agree that the structure alone would not provide enough evidence. However, we provide strong supporting evidence using HDX Mass spectrometry to demonstrate that the presence of 13 -carotene resulted in characteristic structural arrangements. Hence, HDX-MS confirmed the structural changes (solvent exposure) of the protein regions concerned. Further, 13 -carotene binding was supported by molecular docking (Supplementary Data 14-17 and Supplementary Data 27-28).

Please report in the method section whether the ligands were used in the crystal freezing steps.

Response: Crystals were soaked with 0.5 mM β -carotene before cryo-protecting them in a solution obtained by mixing equal volumes of 50% glycerol and well solution supplemented with 0.5 mM ligand. These steps are now clarified in the methods section.

PDB ID 8J2V: Residues 310-326 is disordered/out of density/no density, but the authors are making important conclusions based the structural rearrangement upon ligand binding (scopoletin), the density of which is also misleading. Also, the suggested helical transition based on the maps is not suggestive as for 8J2S.

Response: The region 310-326 is comparatively stable only in the presence of UDP-2F-glucose/scopoletin and in the absence of retinol (structure 8J2T -[#]9J9K), as well as in the UDP-bound structure 8J2U. However, we clarify that only the region 310-320 was well defined in the 1 sigma omit map, as now shown in the supporting omit maps (Supplementary Data 13A,B). Residues 321-326 in 8J2T (-[#]9J9K) are less well defined (Supplementary Data 13A,B).

We further acknowledge that the manuscript initially misstated that the scopoletin-only structure is in the closed state. This sentence has now been removed to correct the text. However, once we add retinol to the crystals that were in the closed conformation (as in 8J2T -[#]9J9K and 8J2U), the “closing loop” (between residues 309 and 327) becomes completely disordered and invisible in the electron density maps (8J2S -[#]9J9C) (Supplementary Data 13A,B, Fig. 2).

Please generate omit maps for ligands. Also, the placement of sulfate ion (A/502) is not supported by the density.

Response: Omit maps for all ligands are included (Supplementary Data 7C, Supplementary Data 12A-G; Supplementary Data 13C-E). We replaced the sulfate molecule by a glycerol molecule (glycerol was used for cryoprotection) and deposited in the PDB (8J2V).

PDB ID 8J2T (-[#]9J9K): It is critical to provide omit maps for the ligand density.

Response: Omit maps are included for all ligands in the revised manuscript (Supplementary Data 13C-E).

There is an unaccounted positive density in the active site next to Q353, which seems to be the ligand density.

Response: The density is only visible as positive FoFc, without supporting 2FoFc. Also the distances between the atoms in this region are too small to allow for additional compounds. Hence we think it is only an artifact of the FoFc map.

Also, the rotamer of for critical residue W350 seems to be incorrect. The authors should revisit the maps to better understand the mechanism.

Response: We revisited all the positions of W350 in the UDP/UDP2FG-containing structures 8J2T (-[#]9J9K) (2 molecules in ASU), 8J2S (-[#]9J9C) (7 molecules in the ASU), and 8J2U (2 molecules). The density for the different W350 in these total of 11 molecules was often not strong and somewhat ambiguous. However, we indeed found that collectively, these data would be compatible with W350 being always in the same rotamer position (“parallel to M349”). We have updated the PDB structures accordingly. This consistent W350 rotamer orientation when bound to UDP is clearly different from the W350 rotamer shown in the 2.0 Å resolution structure bound to scopoletin (8J2V).

Authors should provide omit maps for the ligands from the crystallographic studies.

Response: Omit maps are included for all ligands and key residues in the revised manuscript (Supplementary Data 7, 12, and 13).

Specifically, the resolution for 8J31 is poor with very high R-factors. B-carotene interaction distances from the protein side chains (T141, H139 and Y145) seems to be very high. Authors should provide rational and include the electron density maps to support Sup. Fig. 13 (-[#] Supplementary Data 16).

Response: There is a misunderstanding. The conclusion is based on the docking of the ligand and not based on the crystal structure. The 8J31 structure was obtained for the crystals grown in the presence of 13-carotene, soaked with 13-carotene before freezing the crystals. The distance shown between 13-carotene and His139, Thr141 and Tyr145 (Supplementary Data 16) serves only as an example to provide a possible explanation for the inhibitory effect of xanthophylls.

H18, D118, W350 side chain density should be shown as omit maps for Sup. Fig 10 (□ Supplementary Data 12) as the interaction distance is large and seems to be changing based on ligands. Also, Sup. Fig. 10/VI-F: please check H18 and D118 side-chain conformation is supported by the electron density. Also, authors should provide omit maps for UDP2FG-W350 stacking.

Response: Omit maps are now included for these residues and UDP2FG-W350 stacking (Supplementary Data 12A-G)

Authors should include the crystallization condition. Also, please mention whether the corresponding ligands have been used in the cryoprotective solution or not. The presence or absence of ligands in the crystal freezing stage is critical in this case as the affinity is low and the structural analysis/conformational change would be wrongly interpreted.

Response: Crystallization conditions were already provided as a table (Supplementary Table 1) and the methods are updated with additional information as suggested (page 25). The crystals were soaked with 13-carotene and 25% glycerol was added to it for cryoprotection and then the crystals were cryo-cooled.

The use of software should be properly cited: Phenix, COOT, CCP4, chimera, PyMOL, XDS,

Response: Apologies for these omissions; the citations are now all included.

Low molecular weight contamination is evident in SDS-PAGE gel provided by the authors.

Response: Protein was subjected to additional purification steps and the SDS-PAGE gel for it is now included in the manuscript (Supplementary Data 38B). The low molecular weight impurity is GST, which has no effect on the kinetics.

The sample purity is critical for ITC and CD and other biophysical experiments, I recommend to further purify protein sample and redo ITC and CD experiments. The ITC isotherm for buffer control is not ideal and should be repeated along with experimental titration to obtain reasonable ITC isotherm.

Response: The experiments were repeated using the purified protein and updated in the manuscript (Supplementary Data 9 and 20 for CD and ITC, respectively).

Reviewer #2 (Remarks to the Author):

In the submitted manuscript, Liao et al. study the mechanism of substrate inhibition on the enzyme NbUGT72AY1, a glycosyltransferase from *Nicotiana benthamiana*. The authors took advantage of their finding that retinol and 13-carotene, while acting as a competitive inhibitor of the acceptor substrate, scopoletin, appeared to strongly attenuate substrate inhibition resulting from excess acceptor substrate. The authors were able to co-crystallize the enzyme with or without substrates or products in the presence or absence of effectors (including retinol) and obtain X-ray crystal structures that they associated with productive or non-productive complexes of the enzyme. From these structure, the authors propose a mechanistic model to explain substrate inhibition in NbUGT72AY1, in which productive complex formation occurs when the donor substrate (UDP-glucose) binds first followed by the acceptor substrate (scopoletin), whereas a non-productive complex forms when the acceptor substrate binds first followed by the donor substrate. The authors propose that effectors like retinol and 13-carotene can block the acceptor-binding site with minimal conformational change of the protein until the donor can bind and then the effector can be competitively replaced by the acceptor substrate.

As it is estimated that substrate inhibition occurs in roughly 20% of enzymes, insights into substrate inhibition are generally important in enzymology, and of particular interest to industrial biocatalysis and pharmacology with respect to enzyme targets.

The authors state that to date, no structural evidence supports current models of unproductive enzyme-substrate complexes being involved in substrate inhibition, and thus the work reported in this manuscript is important in providing such evidence. (The author's citation of references 1-3 for this lack of evidence refer to some papers that are over 10 years old, however, and the most recent citation is a commentary.)

Response: Reviewer 2 asks for recent references to support the statement that there is no structural evidence to date for unproductive enzyme-substrate complexes that can explain substrate inhibition. References 1-3 are review articles in which various models are presented that attempt to explain substrate inhibition. More recent review articles are not available. However, the mechanism of substrate inhibition is still not clearly understood at the molecular level, as recent original publications show. References that are more recent and have been included in the new version are: 1) Zhang et al., Frustration and the Kinetic Repartitioning Mechanism of Substrate Inhibition in Enzyme Catalysis, *J. Phys. Chem. B* 2022, 126, 6792–6801; 2) Scheerer et al., Allosteric communication between ligand binding domains modulates substrate inhibition in adenylate kinase, *PNAS* 2023 Vol. 120 No. 18 e2219855120; and 3) Vallejos-Bacelliere et al., Characterisation of kinetics, substrate inhibition and product activation by AMP of bifunctional ADP-dependent glucokinase/phosphofructokinase from *Methanococcus maripaludis*, *FEBS Journal* 289 (2022) 7519–7536.

It's not clear to this reviewer, considering that retinol and 13-carotene are competitive with scopoletin (and should occupy to the same acceptor-binding site), why incubation with excess effectors does not also lead to a non-productive complex when they bind before the donor. Is it because these effectors do not cause as large a conformational change as scopoletin does when it binds first, or is it because these effectors do not bind as tightly? The lack of X-ray crystal structures including either retinol or 13-carotene as a ligand leaves this ambiguous.

Response: As Reviewer 2 correctly points out, retinol, 13-carotene and apocarotenoids act as competitive inhibitors when used in high concentrations. Competitive inhibition can be deduced from Fig. 1C, Supplementary Data 33 A and the newly added Supplementary Data 17. As can be seen, the effectors inhibit the enzymatic activity of NBUGT72AY1 when used at high concentrations, but activate it at low concentrations. Therefore, the effectors override the substrate inhibition mechanisms only in a certain concentration range.

Page 11, line 20: I do not agree that the lack of observation of scopoletin in more than one binding site in the X-ray crystal structures refutes the hypothesis that it can bind in more than one site.

Response: Page 11, line 20: With regard to the invisibility of β -carotene and retinol in the NBUGT72AY1 crystal structures, the sentence is amended as follows: No more than one scopoletin molecule could be detected in any of the crystal structures obtained, which makes the hypothesis of a second binding site of the acceptor molecule as the cause of SI unlikely.

Overall, the work is well done. Methodology is sound and enough detail is provided.

Reviewer #3 (Remarks to the Author):

This article proposes a molecular mechanism of substrate inhibition of a tobacco glycotransferase. The authors utilised x-ray crystallography to offer snapshots of the molecular reaction mechanism involving a bi-substrate enzyme. The authors also employ HDX-MS and molecular dynamics to shed light on the conformational changes occurring substrate binding in active and inhibited protein complexes. Overall, this appears to be a comprehensive study that aims at explaining substrate inhibition in multi-substrate enzyme. However, I feel that the study can be improved following some suggestions below:

1. Whilst HDX-MS constitutes a small part of the article, the presentation of data lacks some information. There is no information about statistics of the experiments. For instance how many technical and biological replicates have been performed? In supp figures 7 and 22, it is hard to judge whether or not the differences in deuterium uptake are statistically significant or not. I would suggest the authors to use the wood's plots. There are specialised softwares (e.g. Deuterios) that can carry out such analyses.

Response: Supplementary Table 6 detailing the HDX-MS experiments was amended with the above-mentioned information (Overview) in line with the recommendations by Masson et al. 2019. Time course studies were carried out (Supplementary Table 6). Three technical replicates (separate H/D exchange reactions) were analyzed.

We agree that the conventional heat maps displaying HDX differences along the amino acid sequence lack information about significance of individual peptides, and that Woods plots would contain that information. However, Woods plots on the other hand have the downside of only showing one HDX incubation time point thereby neglecting dynamic behaviour over the whole time-course.

We hope that it is acceptable for the reviewer that we thus stick to showing heat maps of HDX changes. In these we had set the threshold of HDX changes to $\pm 6.25\%$ difference, which when being exceeded represents changes in blue color. With the statistics provided in the amended Supplementary Table (as average repeatability) these represent significant changes.

2. What is the back exchange in the HDX-MS experiments? From Supp Figures 7 and 22 it appears that HDX differences are small (e.g. from -20 to 20). That may indicate high back exchange (>30%). It will be good if the authors can comment on this.

Response: As the presented HDX-MS experiments were solely comparative, back-exchange controls were not strictly required for interpretability and thus not performed. This is now also explicitly mentioned in the amended Supplementary Table 6 detailing the HDX-MS experiments (overview).

3. I would recommend the authors to include information about the peptide coverage, redundancy they obtained for each experimental condition (see recommendation from the community white paper that the authors cite here, Masson et al). Suppl. Table 6 provides such information, but this needs to be expanded.

Response: Supplementary Table 6 detailing the HDX-MS experiments was amended (Residues_HDX). We emphasize that only peptides identified in each of the investigated conditions with similar characteristics were included in the analysis, hence, the reported values for peptide coverage and redundancy apply to the experiment as a whole.

4. Moreover, the redundancy appears a bit low, especially for a soluble protein. It would be good if the authors provide information regarding the optimisation steps that they took to achieve such sequence coverage.

Response: We agree on that. We had not performed any optimization for that particular experiment as we had investigated the very same protein under similar conditions before (Sun 2023 New Phytol and Liao 2023 Plant Communication). We can thus only relate the relatively poor coverage and redundancy to the compounds investigated in this study, eventually in conjunction as to how we treated the peptides during analyses (see above: only peptides identified in all investigated states were regarded for the analysis).

REPLY TO THE REVIEWERS' COMMENTS

Reviewer #1 (Remarks to the Author):

Although, the authors have made substantial improvement in the revised manuscript, I still have concerns, which needs to be addressed for the manuscript to be considered for publication in Nature Communications.

COMMENT: I'm concerned about the placement of scopoletin in 8J2V. Although, the omit map has been provided, the fitting is not clear as the FoFc density map does not seem to be supporting the full ligand geometry. Authors should check the fitting and perform occupancy refinement and provide clear picture for the occupancy of the ligand.

RESPONSE: We thank the reviewer for their query regarding ligand occupancy. We have adjusted the scopoletin position to better fit the density, and refined the occupancy for the ligand using Phenix_refine, resulting in an occupancy of 0.73. The B-factor for the ligand at this occupancy value is now closer to those of the neighboring protein residues (Supplementary Table 2). The output from the refinement run for the PDB, along with the respective occupancy for the ligand, is provided below. We have also deposited the updated coordinate files to the PDB database for 8J2V.

PDB output after the occupancy refinement (PDB: 8J2V)

HETATM 3629	CAA T83 A 501	20.643	21.521	23.235	0.73	50.49	C
HETATM 3630	CAD T83 A 501	25.784	17.232	22.095	0.73	42.32	C
HETATM 3631	CAE T83 A 501	25.061	18.248	22.785	0.73	42.48	C
HETATM 3632	CAF T83 A 501	21.597	17.334	21.750	0.73	48.91	C
HETATM 3633	CAG T83 A 501	22.847	19.271	23.321	0.73	45.37	C
HETATM 3634	CAJ T83 A 501	20.836	18.313	22.407	0.73	50.25	C
HETATM 3635	CAK T83 A 501	21.463	19.288	23.190	0.73	50.41	C
HETATM 3636	CAL T83 A 501	25.070	16.311	21.347	0.73	48.25	C
HETATM 3637	CAM T83 A 501	23.619	18.267	22.662	0.73	44.19	C
HETATM 3638	CAN T83 A 501	23.026	17.321	21.887	0.73	46.98	C
HETATM 3639	OAB T83 A 501	25.815	15.333	20.689	0.73	45.57	O
HETATM 3640	OAC T83 A 501	19.435	18.340	22.288	0.73	55.51	O
HETATM 3641	OAH T83 A 501	20.704	20.268	23.852	0.73	62.24	O
HETATM 3642	OAI T83 A 501	23.755	16.370	21.252	0.73	51.13	O

COMMENT: Also, it is highly unlikely that the ligand has only two interacting atoms more than 4Å apart. Please also provide the LigPlot for scopoletin.

RESPONSE: We thank the reviewer for their suggestion to include a Ligplot for Scopoletin. We have now included the Ligplot figure in the Supplementary data 7E, which shows the neighboring protein residues.

Ligplot for Scopoletin from PDB: 8J2V

COMMENT: I have attached FoFc map for scopoletin at 0.7 sigma and 0.6 sigma. 2Fo-Fc map at 1 sigma cutoff

RESPONSE: 2Fo-Fc map for Scopoletin at 1 sigma and 0.7 sigma cutoffs is provided below following the aforementioned refinement run (left and middle panel). Additionally, we ran a Phenix polder map for Scopoletin (right panel), and the output below confirms the presence of Scopoletin density, which was omitted during the run.

2Fo-Fc map at 1 sigma cutoff

2Fo-Fc map at 0.71 sigma cutoff

Polder map from Phenix

Polder map output from Phenix:

File Glucosyl_Transferase_refine_228_polder_map_coeffs.mtz was written.

Map 1: calculated Fobs with ligand

Map 2: calculated Fobs without ligand

Map 3: real Fobs data

CC(1,2): 0.7777

CC(1,3): 0.9017

CC(2,3): 0.7636

Peak CC:

CC(1,2): 0.8501

CC(1,3): 0.8874

CC(2,3): 0.8097

q D(1,2) D(1,3) D(2,3)

0.10 0.3900 0.5064 0.6685

0.20 0.3604 0.4102 0.5341

0.30 0.2984 0.3549 0.4341

0.40 0.3058 0.3238 0.4017

0.50 0.3209 0.2890 0.3437

0.60 0.3532 0.2811 0.3418

0.70 0.3907 0.2214 0.3516

0.80 0.4074 0.2336 0.3703

0.90 0.4862 0.2861 0.4684

0.91 0.5009 0.2922 0.5148

0.92 0.5109 0.2973 0.5233

0.93 0.5356 0.3186 0.5181

0.94 0.5819 0.3637 0.5415

0.95 0.5949 0.3646 0.5278

0.96 0.6588 0.3976 0.5697

0.97 0.6892 0.4386 0.5795

0.98 0.8023 0.5000 0.6046

0.99 0.9438 0.6676 0.6906

The polder map is likely to show the omitted atoms.

COMMENT: The lack of B-carotene (8J31) and model error (residue 53-63) and poor resolution suggest over interpretation. The authors should try rMMS technique to obtain protein crystals in new condition in presence of very high ligand concentration. Without the ligand density, this structure is not telling anything significant.

RESPONSE: We have attempted all possible methods, including the random matrix microseeding technique and using the maximum feasible concentration of beta-carotene despite its poor solubility. Unfortunately, the ligand density is still not visible in the structure. However, other biophysical methods (CD, HDX-MS) and kinetic data confirm its interaction with the protein. Conclusions about the mechanism of *NbUGT72AY1* are not derived from individual structures or methods in the manuscript, but by group comparisons. Although it was not possible to resolve the β -carotene structure in 8J31, the kinetic data, HDX data and CD spectra support the interaction with the protein. Despite its shortcomings, 8J31 lends supports to the proposed mechanism. No conclusion is drawn from the structure of 8J31 that is not backed up by other methods.

COMMENT: There are several Ramachandran outliers and clash score is very high in 9J9C. CC1/2 and I/sigI is very low as compared to other structures. Authors should improve the structure quality. Authors should take advantage of alphafold model.

RESPONSE: We have corrected the Ramachandran outliers and reduced the clash scores for PDB 9J9C based on manual inspection and MolProbity outputs. The updated structure has been deposited to the PDB database. Additionally, we improved the CC1/2 and I/sigI for the structure by reprocessing the data using Staraniso (<https://staraniso.globalphasing.org/cgi-bin/staraniso.cgi>), resulting in a CC1/2 of 0.39 and an I/sigI of 1.9 in the outer shell (Supplementary Table 2).

Molprobity outputs for the previous version:

Clashscore, all atoms:		43.3
Ramachandran outliers	82	2.57%

Molprobity outputs for the current version:

Clashscore, all atoms:		20.43
Ramachandran outliers	2	0.07%

COMMENT: Authors should pay attention in Table 1. The units are missing at several places. Also, the Ligand B-factor should be included.

RESPONSE: We have updated the missing units and ligand B-factors in Supplementary Table 2.

COMMENT: Please mention the reference model and parameters used in Lorestr refinement.

RESPONSE: We have updated the reference model and parameters used in Lorestr refinement in the methods section as below.

The low resolution structure (9J9C; 3.10Å) was additionally refined using the lorestr pipeline in ccp4, which inspects the data for twinning, checks for the best scaling option to implement for refinement, and uses H-bond external restraints information obtained from homologous structures (PDB: 8J2V and 8J2Z) using proSMART. The refinement incorporates homology-derived restraints to improve the geometry of the model, followed by jelly body refinement. The pipeline evaluates both standard and least-squares scaling methods, opting for the one that yields a lower Rfree value. It also determines the parameters for modeling the solvent, such as VDWProb 1.10, IONProb 0.80, and RSHRink 0.80.

COMMENT: PYMOL should be replaced with PyMOL.

RESPONSE: We have updated the text.

Reviewer #2 (Remarks to the Author):

Comment: All of my comments have been addressed in the revision.

RESPONSE: Thank you for reviewing our manuscript and for your positive response.

Reviewer #3 (Remarks to the Author):

The authors have carried out a poor review of the article barely addressing any comments from HDX-MS experiments. They could have included the wood's plots for all time points used. Also, without assurances about the back-exchanges which appears to be large and lack of optimisation it is still questionable if the data are fully usable. Unless these are done, I cannot recommend publication of the article.

We respectfully disagree with the notion that we barely addressed the reviewer's comments. In fact, the only point we acknowledge not having amended in our previous response was the implementation of Wood's plots.

1) We had argued, and still do, that Wood's plots are a good way of presenting HDX-MS data on the peptide level with the downside that the HDX differences over the whole time-course of an experiment are less apparent and that such visual representation of the HDX kinetics itself is of value. Furthermore, Wood's plots do not make use of overlapping peptides, which in favorable cases allows for narrowing down the area where HDX changes occur to sections smaller than the actual length of the peptides; such an approach is implemented by the heat maps exported from the HDX-MS analysis software DynamX and presented in Supplementary Data 27-28.

Nevertheless, in this revised version of the manuscript, we implemented Wood's plots for both absolute and fractional HDX differences (generated with HaDeX) in the spreadsheet „Supplementary Table 6 HDX-MS“ along with a separate graphical representation of the peptide coverage (see also Supplementary Data 30 to 37). Comparing these to the heat maps presented in the supplementary information of the manuscript, and within the caveat that the former represents *per peptide* comparison while the latter is a *per residue* comparison (see above), we confide that the threshold set by us is rather conservative and minimizes any potential of data over-interpretation.

2) We did not perform specific optimization steps for the dataset at hand as we had investigated the same protein by HDX-MS together with different ligands before (Sun et al. 2023 New Phytol and Liao et al. 2023 Plant Commun). The unanimous relatively poor peptide coverage and redundancy of the dataset at hand may be related to the compounds investigated in the current study, eventually in conjunction as to how we treated the peptides during analyses (only peptides identified in all investigated states were regarded for the analysis as common in such comparative HDX-MS analyses). However, we feel that the presented dataset still contains a sufficient number of peptides to be of value.

3) With respect to the lack of back-exchange controls we would like to emphasize that the guidelines for HDX-MS experiments (Masson GR et al. 2019 Nature Methods) state that „*These [back-exchange] controls are not strictly necessary for comparative measurements between different states of the same protein (for example, with and without ligand), as the back-exchange can reasonably be expected to be the same with each measurement.*“. Exactly such kind of HDX-MS experiment was conducted in this study. Supplementary Table 6 explicitly mentions that no back-exchange control was carried out. Along this line, the highest observable fractional HDX in the dataset at hand was approx. 69% with the D₂O content during the labeling reaction being 86.4%

(see Supplementary Table 6). Hence, the level of back-exchange for these peptides would be approx. 25% while higher back-exchange can be anticipated, for example, for peptides with longer retention times or less favorable amino acid composition, both of which promoting back-exchange. However, back-exchange of up to 50% would be acceptable in bottom-up HDX-MS (see Masson GR et al., 2019, Nature Methods). Related to that subject we disagree that a problem with back-exchange in our experimental setup would be the reason for low HDX difference among the investigated states – HDX changes upon ligand binding may be of small magnitude in cases where only minor conformational changes affecting the amide backbone protons occur.